# The evolution of mechanisms to produce phenotypic heterogeneity in microorganisms

Guy Alexander Cooper [1,2✉], Ming Liu[2], Jorge Peña [3] & Stuart Andrew West [2]

In bacteria and other microorganisms, the cells within a population often show extreme phenotypic variation. Different species use different mechanisms to determine how distinct phenotypes are allocated between individuals, including coordinated, random, and genetic determination. However, it is not clear if this diversity in mechanisms is adaptive—arising because different mechanisms are favoured in different environments—or is merely the result of non-adaptive artifacts of evolution. We use theoretical models to analyse the relative advantages of the two dominant mechanisms to divide labour between reproductives and helpers in microorganisms. We show that coordinated specialisation is more likely to evolve over random specialisation in well-mixed groups when: (i) social groups are small; (ii) helping is more "essential"; and (iii) there is a low metabolic cost to coordination. We find analogous results when we allow for spatial structure with a more detailed model of cellular filaments. More generally, this work shows how diversity in the mechanisms to produce phenotypic heterogeneity could have arisen as adaptations to different environments.

[1] St. John's College, Oxford OX1 3JP, UK. [2] Department of Zoology, University of Oxford, Oxford OX1 3SZ, UK. [3] Institute for Advanced Study in Toulouse, University of Toulouse Capitole, 31080 Toulouse, Cedex 6, France. ✉email: guy.cooper@zoo.ox.ac.uk

Different species use different mechanisms to produce adaptive phenotypic heterogeneity (Fig. 1)[1–5]. In some cases, there is coordination across individuals to determine which individual will perform which role (*coordinated specialisation*)[1,6]. This coordination could use signals, cues, or a developmental programme to provide information about the phenotypes adopted by other individuals in the group[7]. For example, when honey bee workers feed royal jelly to larvae to produce reproductive queens (Fig. 1a), or when the local density of a signalling molecule determines whether cyanobacteria cells develop into sterile nitrogen-fixing heterocysts (Fig. 1b)[8–10]. In other cases, each individual adopts a helper phenotype with a certain probability, independently and without knowledge of the phenotypes adopted by other individuals (*random specialisation*)[2,5,11,12]. For example, in *Salmonella enterica* co-infections, random biochemical fluctuations within each cell's cytoplasm are used to determine whether the cell sacrifices itself to trigger an inflammatory response that eliminates competitor species (Fig. 1d)[12,13]. In yet further cases, the phenotype is influenced by the individual's genotype (genetic control). For instance, in some ant societies, whether individuals develop into queens, major or minor workers can be determined, in part, by their genes (Fig. 1c)[3,14–16]. Across the tree of life, some species employ one mechanism to produce phenotypic heterogeneity whereas in other species mixed forms exist with a combination of coordinated specialisation, random specialisation, or genetic control[3,15,17–22].

We lack general evolutionary explanations for why different species use different mechanisms to produce phenotypic heterogeneity[2,3,23,24]. Previous work has focused on the non-reproductive division of labour in the social insects, and the proximate mechanisms that lead to different worker castes[6,16,25–29]. However, the focus in that literature is on a different question—how different proximate mechanisms can produce coordinated specialisation—rather than the broader question of whether coordinated specialisation should be favoured over random specialisation or genetic control in the first place. It is with the reproductive division of labour that these three very different mechanisms have been observed in different species and for which there is an absence of evolutionary explanations[2,3,23,24,30].

Reproductive division of labour in bacteria and other microbes offers an excellent opportunity for studying why different mechanisms to produce phenotypic heterogeneity are favoured in different species[1,2]. Reproductive division of labour occurs when social groups are composed of more cooperative 'helpers' who gain indirect fitness benefits by the aid they provide to less cooperative 'reproductives'. Across microbes, the two primary mechanisms used to produce reproductive division of labour are coordinated and random specialisation (Fig. 2). Furthermore, while the form of cooperation and life histories of microbes share many similarities, they also vary in factors that could influence the evolution of division of labour, such as social group size[31,32].

We develop theoretical models to examine whether the relative advantages of random and coordinated specialisation can depend upon social or environmental conditions. Our aim is to use the reproductive division of labour in microbes as a 'test system' to address the broader question of whether evolutionary models can explain the diversity in the mechanisms that produce phenotypic heterogeneity more broadly. We show that coordinated specialisation is more likely to evolve over random specialisation in well-mixed groups when: (i) social groups are small; (ii) helping is more "essential"; and (iii) there is a low metabolic cost to coordination. We find the same qualitative results with deliberately simple models that are designed to capture the essence of the problem and with more detailed models that allow for spatial structure.

## Results

We compare the relative fitness advantages of reproductive division of labour with either coordinated or random specialisation. Our first aim is to capture the problem in a deliberately

**a**  Coordinated specialisation    **b**  Coordinated specialisation

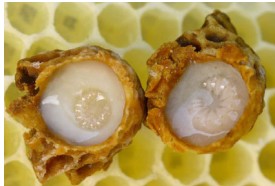    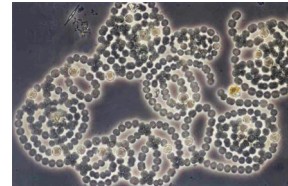

**c**  Coordinated specialisation    **d**  Random specialisation
   + genetic control

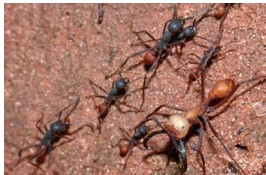    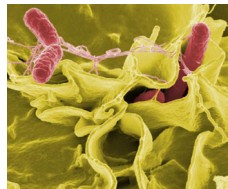

**Fig. 1 Different mechanisms to produce phenotypic heterogeneity in nature. a** In honey bee hives (*Apis mellifera*), larvae develop as sterile workers unless they are fed large amounts of royal jelly by adult workers (coordinated specialisation)[8] (Photo by Waugsberg via the Wikimedia Commons). **b** In cyanobacteria filaments (*A. circinalis*), some individuals develop into sterile nitrogen fixers (lighter/yellow, round cells) if the amount of nitrogen fixed by their neighbours is insufficient (coordinated specialisation). This leads to a precise allocation of labour, with nitrogen-fixing cells distributed at fixed intervals along the filament[9] (Picture by Dr. Imre Oldal via the Wikimedia Commons, cropped). **c** In the army ant (*Eciton Burchelli*), whether individual ants become a major or minor worker has a genetic component (genetic control)[16] (Photo by Alex Wild via the Wikimedia Commons, cropped.). **d** In *S. enterica* infections (serovar Typhymurium), each cell amplifies intracellular noise to determine whether it will self-sacrifice and trigger an inflammatory response that eliminates competing strains (random specialisation)[13] (Photo by Rocky Mountain Laboratories, NIAID NIH via Wikimedia Commons).

**a**  Random specialisation    **b**  Fully coordinated specialisation

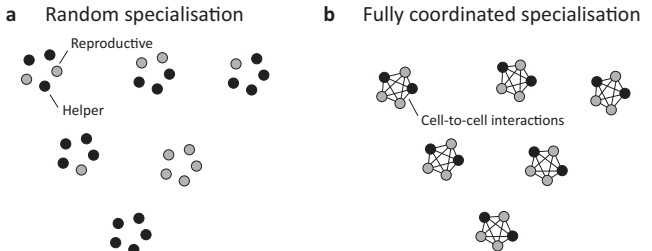

**Fig. 2 Mechanisms to produce reproductive division of labour in clonal groups.** We examine the relative advantages and disadvantages of the two key mechanisms to produce reproductive division of labour in social microorganisms[1,5,11,46]. **a** Random specialisation occurs when cells randomly specialise into helpers or reproductives independently of one another. This can occur when a genetic feedback circuit is used to amplify small molecular fluctuations in the cytoplasm of each cell (phenotypic noise)[4,11,12,78–80]. **b** Coordinated specialisation occurs when cells interact with one another, and share (or gain) phenotypic information while they are differentiating. This could occur through the secretion and detection of extracellular molecules (signals or cues), or with a shared developmental programme (epigenetics)[1,2,25].

simple model, which is easy to interpret, and can be applied across diverse microbe species[33,34].

We begin by assuming that coordinated specialisation always produces the optimal proportion of helpers and reproductives (fully coordinated specialisation) and that there is no within-group spatial structure (well-mixed groups; Methods: Well-mixed groups). We then test the robustness of our results by examining several alternate models for different biological scenarios (Supplementary Methods A–C) and by developing a more detailed model of growing cyanobacteria filaments that include the effects of within-group spatial structure (Methods: Cyanobacteria filaments. Throughout, we assume a form of cooperation that is common in microbes, where some individuals produce a 'public good' that benefits other cells[1,2,9,12,21,35–40].

**Random specialisation vs fully coordinated specialisation**. We assume that a single cell arrives on an empty patch and, through a fixed series of replications, produces a clonal group of $n$ individuals that consists of $k$ sterile helpers and $n - k$ pure reproductives ($k \in \{0, 1, 2, \ldots, n\}$). We define the average group fecundity, $g_{k,n}$, as the reproductive success of a particular group in the absence of mechanism costs. This is measured as the per cell number of offspring that would disperse at the end of the group life cycle, given by

$$g_{k,n} = \frac{1}{n}(n - k)f_{k,n}, \tag{1}$$

where $n - k$ is the number of reproductives in the group, and $f_{k,n}$ is the fecundity of each reproductive in the absence of mechanism costs (Methods: Labour dividers and their fitness). We assume that $f_{k,n}$ increases with the number of helpers in the whole group ($k$).

Expression (1) highlights the trade-off between the number of reproductives in the group ($n - k$), which is higher when there are fewer helpers (lower $k$), and the amount of help that those reproductives obtain ($f_{k,n}$), which is higher when there are more helpers (higher $k$). If the division of labour is favoured, the balance of this trade-off leads to an optimal number of helpers, $k^*$, that is intermediate (i.e., $0 < k^* < n$), giving $g_{k^*,n}$ as the maximal reproductive success of the group (Methods: Fully coordinated specialisation).

In species that divide labour by coordination, the outcome of individual specialisation depends on the phenotypes of social group neighbours. Our first model is deliberately agnostic to the details of how phenotype information is shared between group members in order to facilitate predictions across different systems. For instance, individuals may share phenotype information via signalling between cells or with a common developmental programme (Fig. 2b)[1,2,41]. We make the simplifying assumption that individuals coordinate fully so that coordinated groups always form with precisely the optimal number of helpers, $k^*$. The disadvantage of coordinated specialisation is that the mechanism could incur metabolic costs, such as the production of extracellular signalling molecules. The fitness of a group of coordinated specialisers is given by:

$$w_C = (1 - c_C)g_{k^*,n}, \tag{2}$$

where $g_{k^*,n}$ is the average group fecundity with the optimal number of helpers, $k^*$, and $0 \leq c_C \leq 1$ is the metabolic cost of coordination, whose form we leave unspecified but could in principle depend on further factors such as group size (Methods: Fully coordinated specialisation). We use 'metabolic costs' as a shorthand for all fixed costs at the time of differentiation, which could include other factors such as delaying reproduction. A number of different models have examined how different

proximate mechanisms can produce coordinated division of labour in specific systems[6,25,28,29].

In species that divide labour by random specialisation, each individual in the group independently becomes a helper with a given probability and a reproductive otherwise (Fig. 2a). Hence, the final number of helpers in the group is a binomial random variable. We assume here that the probability of adopting a helper role is equal to the optimal proportion of helpers ($p^* = k^*/n$). In principle, differences between the optimal probability of adopting a helper role and the optimal proportion of helpers could arise if there are different costs on average from producing groups with more or fewer helpers than is optimal. In Supplementary Methods A.2, we show that the same qualitative results arise if the probability of adopting a helper role maximises the fitness of randomly specialising cells. Thus, the expected fitness of a group of random specialisers is given by:

$$w_R = (1 - c_R)\sum_{k=0}^{n}\binom{n}{k}p^{*k}(1 - p^*)^{n-k}g_{k,n}, \tag{3}$$

where $0 \leq c_R \leq 1$ is the metabolic cost of random specialisation, which we assume is independent of the number of helpers in the group, $k$ (Methods: Random specialisation). The potential advantage of random specialisation is that there may be fewer upfront metabolic costs from, for example, between cell signalling (i.e., if $c_R < c_C$ holds). The downside of random specialisation is that it can incur a stochastic cost: groups will often form with fewer or more helpers than is optimal (developmental stochasticity). Stochastic costs occur whenever groups arise with a sub-optimal composition, which is captured in our model by how the fitness of the group depends upon the number of helpers (Eq. 3).

The optimal proportion of helpers could depend upon the environment, and the probability of becoming a helper could be conditionally regulated in response to environmental cues. However, for simplicity, all our analyses assume a stable environment and ignore such regulation.

We need to specify how reproductive fecundity depends on the number of helpers in the group. This relationship will determine the functional cost of having a sub-optimal proportion of helpers. We focus on one of the most common forms of cooperation in microbes, where individuals secrete factors that provide a benefit to the local population of cells ("public goods")[38]. We assume that the amount of public good in the social group depends linearly on the number of helpers in the group and is "consumed" by all group members equally[42,43]. An example of such public good is found in *Bacillus subtilis* populations, where only a subset of cells (helpers) produce and secrete proteases that degrade proteins into smaller peptides, but where these are then re-absorbed as a nutrient source by all cells[44,45]. Further experimental evidence is needed to show that the non-helper cells in *B. subtilis* populations are more reproductive.

We allow the relative importance of producing public goods to vary between species. Each reproductive has a baseline fecundity, $b \geq 0$, that is independent of the amount of public good in the group. The fecundity benefit of helpers scales according to $h \geq 0$ as the amount of public good in the group increases. When reproductives have no baseline fecundity ($b = 0$) we say that cooperation is essential. When baseline fecundity is non-zero ($b > 0$), cooperation is non-essential and the ratio $h/b$ provides a useful metric for the relative importance of cooperation.

Our assumptions give the following expression for the fecundity of a reproductive (Methods: Linear public goods):

$$f_{k,n} = b + h\frac{k}{n}, \tag{4}$$

By substituting Eq. 4 into Eqs. 1–3, we can determine when the fitness of coordinated specialisation is greater than the fitness of

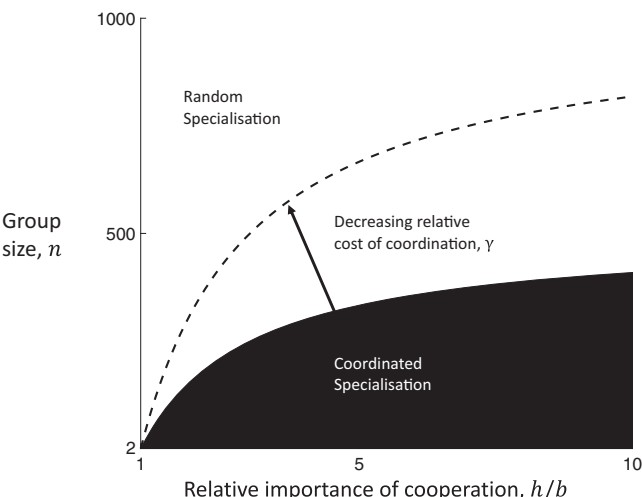

**Fig. 3 Random versus coordinated specialisation.** Small group sizes (lower $n$), relatively more important cooperation (higher $h/b$), and lower relative metabolic costs to coordination (lower $\gamma$) favour division of labour by coordinated specialisation (black) over division of labour by random specialisation (white). This is a visual depiction of Condition 5. We have used $\gamma = 2 \times 10^{-3}$ (solid boundary) and $\gamma = 1 \times 10^{-3}$ (dashed boundary). We note that the limit as the relative importance of cooperation goes to infinity (very large $h/b$) converges to the outcome for when cooperation is essential ($b = 0$).

random specialisation (i.e., $w_C > w_R$), which gives the simplified condition:

$$\gamma < \frac{h - b}{n(h + b)}, \qquad (5)$$

where $\gamma = (c_C - c_R)/(1 - c_R)$ captures the relative change in metabolic costs paid when switching to coordinated specialisation from random specialisation (Methods: Linear public goods). If $h < b$, then sterile helpers are disadvantageous and the group is composed of all reproductives ($k^* = 0$). Thus, division of labour with sterile helpers is favoured to evolve only when $h > b$, which we will assume henceforth. In order for coordination to be favoured, the relative metabolic cost of coordination, $\gamma$, must be less than the relative fitness advantage of coordination over random specialisation in the absence of metabolic costs. Consequently, we term the right-hand side of Eq. 5 as the stochastic cost of random specialisation (Supplementary Discussion). Thus, Condition (5) specifies that coordinated specialisation is favoured when the relative metabolic cost of coordination ($\gamma$), is less than the stochastic cost of random specialisation (right-hand side). The condition can be used to predict how key environmental and ecological factors will influence which labour-dividing mechanism is more likely to evolve (Fig. 3).

**Prediction 1.** Smaller relative metabolic costs of coordination favour coordinated specialisation. When the metabolic cost of coordination is smaller (lower $c_C$) and the metabolic cost of random specialisation is larger (higher $c_R$), then the relative cost of switching from random specialisation to coordinated specialisation is lessened (smaller $\gamma$), which favours the evolution of coordinated specialisation (smaller left-hand side of Condition 5). If the metabolic costs of random specialisation are equal to or larger than the metabolic costs of coordination ($c_R \geq c_C \Rightarrow \gamma \leq 0$), then coordinated specialisation is always the favoured mechanism (Condition 5 always satisfied). Conversely, random specialisation can only ever be the favoured strategy ($w_R > w_C$; Condition 5 not satisfied) if the metabolic costs of random specialisation are less

than the metabolic costs of coordination ($c_C > c_R \Rightarrow \gamma > 0$; a necessary but not sufficient condition). This arises directly from our starting assumption that coordinated specialisation always produces groups with the optimal proportion of helpers whereas random specialisation may often produce groups that are sub-optimal.

Larger metabolic costs of coordinated specialisation ($c_R < c_C \Rightarrow \gamma > 0$) may be a reasonable assumption for many biological systems. The metabolic costs of random specialisation are determined by the production costs of the regulatory proteins employed in the genetic feedback circuit that amplifies intracellular noise[4,5,46,47]. In contrast, coordinated specialisation requires both an intracellular genetic feedback circuit and some mechanism by which phenotype is communicated between cells, such as the costly production and secretion of extracellular signalling molecules[1,2,9,41,48,49]. Coordinated specialisation could also take more time, leading to delayed reproduction.

If coordination is more metabolically costly ($c_C > c_R$), the optimal mechanism to divide labour depends on how the relative metabolic cost of coordination ($\gamma > 0$) balances against the benefit of avoiding the stochastic cost of random specialisation (right-hand side of Condition 5). The stochastic cost of random specialisation is determined entirely by: (i) the relative likelihood that random groups deviate from the optimal proportion of helpers, and (ii) the degree to which those deviations from the optimal proportion of helpers leads to a reduced average fecundity for the group (Methods: Linear public goods). Equation (5) shows how the importance of these two factors depends upon the size of the group ($n$) and on the relative importance of cooperation ($h/b$).

**Prediction 2.** Smaller social groups favour coordinated specialisation. The number of cells in the group has a large impact on the relative likelihood that random groups deviate from the optimal proportion of helpers (Fig. 3). In smaller groups, random specialisation can lead more easily to the formation of groups with a realised proportion of helpers that deviates significantly from the optimum. In contrast, in larger groups, the realised proportion of helpers will be more closely clustered about the optimal proportion with the highest fitness. This effect of group size on the stochastic cost of random specialisation is a consequence of the law of large numbers. For example, outcomes close to 50% heads are much more likely when tossing 100 coins in a row compared to only tossing 4 coins in a row where no heads or all heads may frequently occur.

Our prediction is related to a previous result from sex allocation theory. When mating occurs in small groups, small brood sizes select for more precise and less female-biased sex ratios, as there would otherwise be a high probability of producing a group containing no males at all[50–52]. In another analogue, Wahl showed a mechanistically different effect when the division of labour is determined genetically and the number of group founders is small: groups may sometimes form that do not contain all of the genotypes required to produce all of the necessary phenotypes in the division of labour[24].

**Prediction 3.** The higher the relative importance of cooperation, the more coordinated specialisation is favoured. When the relative importance of cooperation is larger (higher $h/b$), the fitness costs incurred from producing too few helpers increases. In addition, as the relative importance of cooperation increases (higher $h/b$), the optimal proportion of helpers increases to 50% helpers ($p^* \approx \frac{1}{2}$). This increases the variance in the proportion of helpers produced by random specialisers, and so sub-optimal groups may arise even more frequently (Methods: Linear public goods). Thus, higher

relative importance of cooperation increases both: (i) the likelihood that groups deviate from the optimal proportion of helpers; and (ii) the scale of the fitness cost when they do. Both of these effects increase the stochastic cost of random specialisation (larger right-hand side of Condition 5), and thus favour the evolution of coordinated specialisation (Fig. 3).

**Alternative forms of cooperation**. The above analysis employs a deliberately simple public goods model, focusing on factors that are expected to be relevant across many microbial systems. This facilitates the interpretation of our results and generates broadly applicable predictions that are less reliant on the details of particular species.

In order to test the robustness of our results (predictions 1–3) we also developed a series of alternative simplified models corresponding to different biological scenarios (Supplementary Methods A and B; Supplementary Figs. 1–3). We examined the possibility that the public good provided by helpers: (i) is not consumed by its beneficiaries, as may occur when self-sacrificing *S. enterica* cells enter the gut to trigger an immune response that eliminates competitors (non-rivalrous or non-congestible collective good; Supplementary Methods B.2); or (ii) is only consumed by the reproductives in the group, as may preferentially occur for the fixed nitrogen secreted by heterocyst cells in *A. cylindrica* filaments (excludible or club good; Supplementary Methods B.3)[9,12,53,54]. We allowed for randomly specialising cells to maximise their own probability of becoming helpers (Eq. 3; Supplementary Methods A.2), for reproductive fecundity to depend non-linearly on the proportion of helpers in the group (Supplementary Methods A.3), for helpers to have some fecundity (non-sterile helpers; Supplementary Methods B.4), and for division of labour to occur in each generation of group growth (Supplementary Methods B.5). In all of these alternative scenarios, we found qualitative agreement across the three predictions of the linear public goods model.

We found that less specialised helpers (with some fecundity) favour random specialisation over coordinated specialisation (Supplementary Methods B.4). In contrast to prediction 3, more fecund helpers can lead to a scenario where the larger relative importance of cooperation (higher $h/b$) disfavours coordinated specialisation. This occurs because high relative importance of cooperation (higher $h/b$) can produce groups composed predominantly of non-sterile helpers ($p^* \approx 1$), where the likelihood that random groups deviate from the optimal proportion of helpers is significantly diminished.

In Supplementary Methods C, we develop an individual-based simulation that also supports predictions 1–3. In addition, this simulation shows that costly coordination can evolve incrementally from random specialisation, and that intermediate levels of coordination can be favoured (Supplementary Fig. 4)[55].

**Division of labour in a cyanobacteria filament**. We then developed a more mechanistically detailed model of a growing cyanobacteria filament to investigate the impact of within-group spatial structure (Methods: Cyanobacteria filaments)[56]. When there is insufficient fixed nitrogen ($N_2$) in the environment, some cyanobacteria species will facultatively divide labour between reproductive cells (autotrophs) that photosynthesise light and sterile helper cells (heterocysts) that fix and secrete environmental $N_2$ (Fig. 1b)[9,57,58]. The fixed $N_2$ diffuses along the filament where it is used by reproductives to grow and produce new cells. Division of labour in cyanobacteria is a canonical example of coordinated specialisation as helpers produce a variety of signalling molecules that diffuse along the filament to ensure that a regular pattern of

phenotypes develops (Fig. 1b)[9,57,58]. Previous models of cyanobacteria focused on determining the signalling and regulatory network required to recreate the exact pattern of heterocysts along the filament[57,59–63].

Cyanobacteria spores (hormogonium) tend to contain multiple cells[9,57]. In order to consider the case where cooperation is essential, we assume that each filament begins as a clonal sequence of two reproductives (R) and two helpers (H) in the arrangement H-R-R-H (Methods: Life cycle). In Methods: Simulation results and Supplementary Fig. 6, we show that the same qualitative results hold for the alternative assumption where all spore cells are reproductive (R-R-R-R). Over time, the number of cells in the filament increases as reproductives grow and divide by binary fission to produce within-filament offspring cells, which become either helpers or reproductives (Fig. 4a). The group life cycle ends when the filament has reached a maximum size of $L$ cells. At this time, the reproductives in the filament produce dispersing spores that found filaments in the next generation of the group life cycle and all remaining cells die (non-overlapping generations)[9].

Reproductives grow over time by absorbing fixed $N_2$, until they reach a critical size for cellular replication (Methods: Size of reproductives and Replication of reproductives). Each reproductive receives fixed $N_2$ from the abiotic environment at a rate of $\phi \geq 0$ units of fixed $N_2$ per unit time (uniform background density of fixed $N_2$)[63]. In addition, each helper in the filament produces fixed $N_2$, at a maximum rate of $\bar{\phi} > 0$ units of fixed $N_2$ per unit time. We assume that the fixed $N_2$ produced by a helper disperses across the filament with a diffusion factor, $0 < \eta \leq 1$, where limited diffusivity (small $\eta$) means that only reproductives near the helper benefit from the fixed $N_2$ it produces and high diffusivity (large $\eta$) means that even distant reproductives along with the filament benefit (Methods: The local density of the public good across the filament). For the purposes of a focused analysis on the reproductive division of labour, we ignore other forms of phenotypic heterogeneity that cyanobacteria filaments may engage in, such as the production of ATP for the group by autotrophs (non-reproductive division of labour) and the formation of persistor cells in some environments (bet-hedging)[1,58,64–66].

Upon replication, whether a new cell becomes a helper or a reproductive depends on four evolutionary traits that jointly determine the extent of division of labour and coordination in the filament ($q, s, d$, and $v$; Fig. 4b; Methods: How cells specialise). The baseline probability ($0 \leq q \leq 1$) is the underlying probability that a cell becomes a helper in the absence of coordination. The level of signalling ($0 \leq s \leq 1$) is the fraction of resources that a helper commits to the production and secretion of signalling molecules. The signalling molecules produced by a helper disperses along the filament with a diffusivity that we assume is distinct from the $N_2$ diffusivity (Fig. 4a). The local density of signalling molecules allows new cells to estimate how close they are to a helper, or how many helpers there may be nearby.

Whether and how the new cell responds to the signal depends on the response sensitivity ($v \geq 0$) and the response threshold ($d \geq 0$; Fig. 4b; Methods: How cells specialise). If $v = 0$, then a new cell is insensitive to the signal and adopts the helper phenotype with the baseline probability $q$ (random specialisation). If the new cell is sensitive to the signal ($v > 0$) then a local signal density that is greater than the response threshold, $d$, will lead the cell to be less likely to adopt the helper phenotype (Fig. 4b). A higher signal density than the threshold produces the opposite effect. As sensitivity increases (higher $v$), the response to the signal becomes more deterministic (Fig. 4b).

Increasing levels of coordination (higher $v$ and $s$), allows for more precise patterning of helpers and reproductives in the

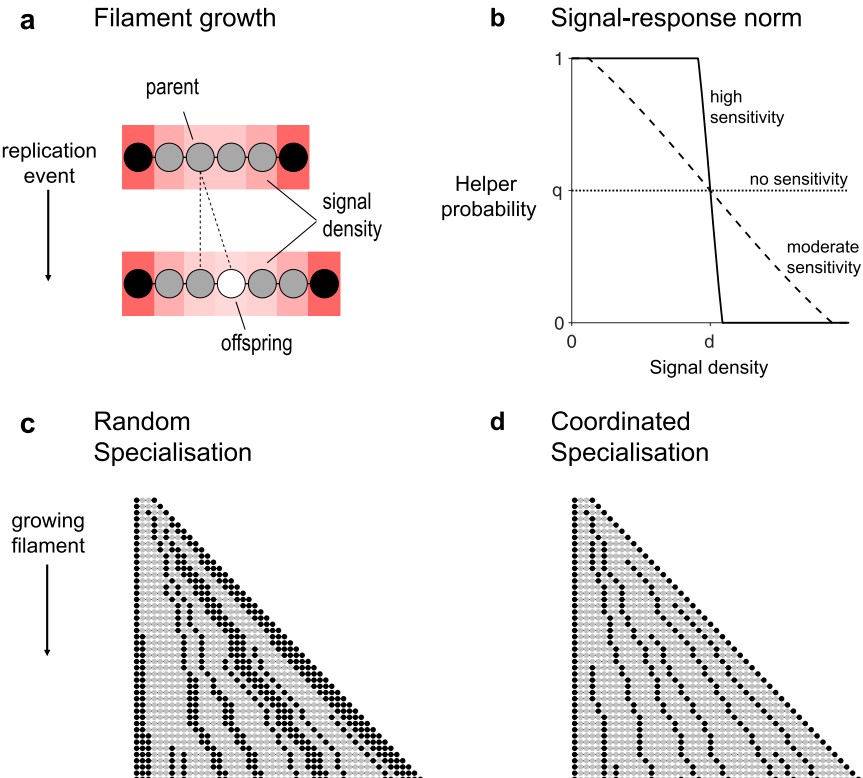

**Fig. 4 Division of labour in a cyanobacteria filament. a** Black cells represent helpers, and grey cells reproductives. When a reproductive replicates, the parent cell produces an offspring cell (white cell) to one side of itself along the filament. The blue shading shows the density of the signal molecule produced by the helpers as it diffuses along the filament. **b** When an offspring cell is sensitive to the signal ($v > 0$), a greater (lesser) signal density will decrease (increase) the probability that it becomes a helper ($q = 0.5, d = 5, v = 0, 0.2, 1$). **c** A simulated example of a filament growing that employs random specialisation ($q = 0.33, s = 0, d = 0,$ and $v = 0$). **d** A simulated example of a filament growing that employs coordinated specialisation ($q = 0.33, s = 0.1, d = 1$ and $v = 1.5$) (Methods: Cyanobacteria filaments). The helper cells (black) are more evenly spaced out (less clumped) with coordinated specialisation compared to with random specialisation.

filament (compare Fig. 4c, d). However, we assume that increased coordination is metabolically costly. First, as helpers produce more signalling molecules (higher $s$), they can produce proportionally less fixed $N_2$. Second, new cells that are more sensitive to the local density of the signalling molecule (higher $v$) incur a more severe time delay before they can specialise, such that reproductives ultimately take longer to reach the critical size of replication.

Cyanobacteria filaments employ such a signalling system and do not simply use the local density of fixed $N_2$ as a cue. A possible reason for this is that signalling molecules could be fast to produce and secrete and thus coordination can occur even before helpers begin to fix $N_2$[62]. Furthermore, using a dedicated signal could be more reliable than one based on fixed $N_2$ density alone, which might be biased by transient fluctuations in the background level of fixed $N_2$ ($\phi$).

**Simulations**. We simulated an evolving population to estimate the strategy that is favoured by natural selection in different scenarios ($q^*, s^*, d^*, v^*$) (Methods: Evolution of coordination and Simulation results, and Supplementary Table). We started with a uniform population that specialises randomly ($s = d = v = 0$), and allowed the helper probability ($q$) to mutate and evolve for 500 generations, until an approximate equilibrium was reached. We then held the baseline helper probability ($q$) fixed and allowed the coordination traits ($s, d$ and $v$) to mutate and evolve for 3500 generations. Each generation, the mutant strategy successfully replaces the resident strategy if it has a higher estimated average fitness. We calculate the fitness of individual filaments as

the summed fecundity of reproductives in the last generation of the group life cycle, divided by the amount of time that it took the filament to grow to L cells. The separate phases of the evolutionary simulation facilitate cleaner convergence of trait values, with an equilibrium generally being reached within 100–200 generations (Supplementary Fig. 5).

We found that the degree to which specialising cells evolve to coordinate can depend on social and environmental factors. In particular, both a lower background density of fixed $N_2$ (small $\phi$) and more limited diffusion of fixed $N_2$ along the filament (smaller $\eta$) lead to the evolution of higher signalling levels (larger $s^*$; Fig. 5a) and higher response sensitivities (larger $v^*$; Fig. 5b). This produced filaments with a more precise allocation of labour across the filament (Fig. 5e). We quantify the extent of coordination by dividing the variance among the number of helpers in a contiguous sub-block of 10 cells by the variance that would be expected for a binomial random variable of the same mean (Methods: Simulation results). Higher values of the reciprocal of this ratio indicate the more precise division of labour.

The predictions of our cyanobacteria model agree broadly with those of our simpler analytical model. When there is limited diffusion of helper-fixed $N_2$ (low $\eta$), reproductives must depend primarily on helpers that are nearer along the filament, producing a smaller effective social group size (analogous to lower $n$). With random specialisation, a smaller social group can lead to proportions of helpers that deviate more from the optimum, increasing the benefit that can be obtained by coordination (Fig. 3). When the background density of fixed $N_2$ is small (low $\phi$), this increases the relative benefit

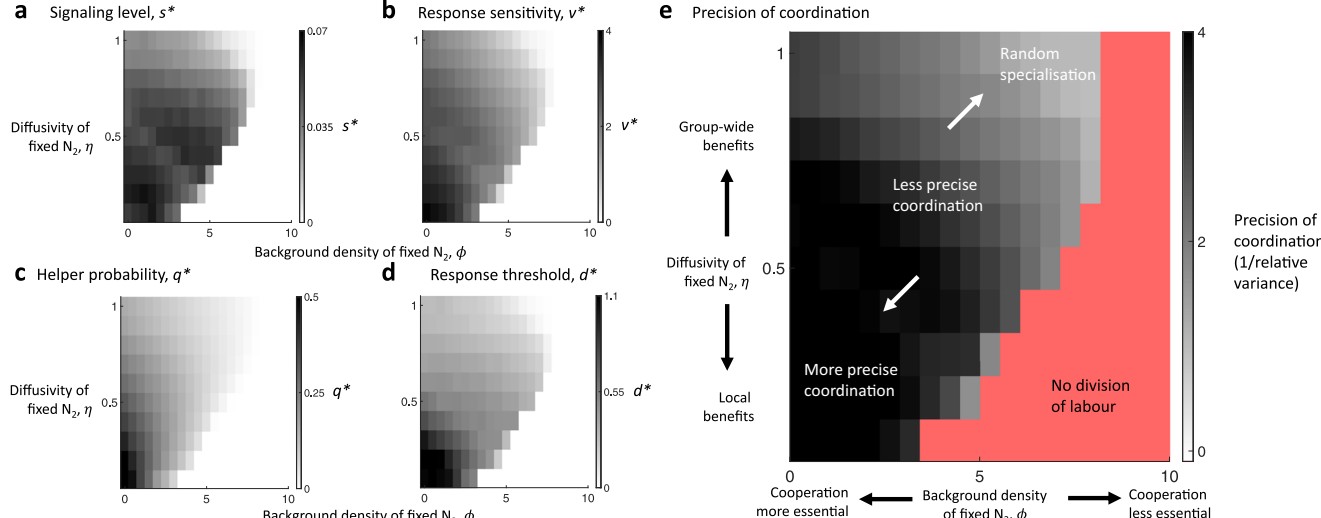

**Fig. 5 The optimal level of coordination.** We present simulation results for two key factors that affect the optimal level of coordination (Methods: Simulation results). A lower background density of fixed $N_2$ (smaller $\phi$) and more limited diffusion of helper-fixed $N_2$ (smaller $\eta$) favours: **a** the evolution of a higher level of signalling (larger $s^*$); **b** a higher response sensitivity to the signal (larger $v^*$); **c** a higher baseline helper probability (larger $q^*$); and **d** a higher response threshold (larger $d^*$); **e** The effect of higher levels of both signalling (larger $s^*$ in **a**) and response sensitivity (larger $v^*$ in **b**) is that groups form with a more precisely coordinated helper distribution. The precision of coordination is calculated by dividing the variance in the number of helpers in a contiguous sub-block of 10 cells relative to the variance that would be expected for a binomial random variable of the same mean (Methods: Simulation results). Higher values of the reciprocal of this ratio suggest more precisely coordinated division of labour (darker shades).

of cooperation (analogous to higher $h/b$). With an increased benefit from cooperation, there is a greater advantage from coordinating to produce the optimum proportion of helpers (Fig. 3). In addition, our cyanobacteria model shows how intermediate coordination can be favoured in certain scenarios (Fig. 5).

However, care is required when examining factors in mechanistic models that can have additional effects unaccounted for by their analogues in simpler models. For instance, an increase in the background density of fixed $N_2$ (higher $\phi$) means that cooperation is relatively less important (lower $h/b$), which we have found favours less coordination (Fig. 5). Relatively less important cooperation (lower $h/b$) in the mechanistic model also means that helpers may be willing to dedicate more effort to signal production (higher s) as there is then a relatively lower fitness cost to producing less of the public good. Another example is how helpers that produce more fixed $N_2$ (larger $\bar{\phi}$) not only lead to cooperation that is relatively more important (higher $h/b$) but can also lead to larger effective social groups sizes (larger $n$) as the increased good that helpers produce can then diffuse further along with the filament and benefit reproductives that are farther away.

**Spatial structure and helper clumping.** Our simulations show that coordination ($s^* > 0$, $v^* > 0$) is often favoured over random specialisation ($s^* \approx 0$, $v^* \approx 0$; Fig. 5a, b). In social groups with rigid spatial structure and local cooperation (lower $\eta$), an effective division of labour requires a regular distribution of helpers across the group. We hypothesised that random specialisation is particularly disadvantageous in such groups because it can lead to contiguous groups of helpers (clumps) that expand as the whole group grows, incurring a high functional or stochastic cost (compare Fig. 4c, d; Supplementary Fig. 7). The helpers within these clumps can neither reproduce to break up the clump nor are they close enough to reproductives to provide fixed $N_2$. We performed additional simulations to investigate the likelihood and impact of helper clumping in growing filaments (Methods: The effect of helper clumping).

We found that a lower background density of fixed $N_2$ (smaller $\phi$) and more limited diffusion of fixed $N_2$ (smaller $\eta$), leads to randomly specialising filaments with a larger average clump size (measured in the number of helpers per clump; Fig. 6a), and a higher cost of clumping (measured as the slope of the best-fit line of average clump size on filament fitness; Fig. 6b). A higher propensity to form clumps arises because a lower background density of fixed $N_2$ (smaller $\phi$) and more limited diffusion of fixed $N_2$ (smaller $\eta$) means new cells are more likely to become helpers (larger $q^*$; Fig. 5c). A higher cost to clumping arises in this case (smaller $\phi$ and $\eta$) because reproductives that are far from helpers have much lower fecundity, which increases the pressure for an even distribution of helpers (high functional costs). Combined, these patterns help to explain why random specialisation is disfavoured in this extreme (lower left corner of Fig. 5a, b, e).

Focusing on the extreme case of essential cooperation ($\phi = 0$) and very low diffusion of fixed $N_2$ ($\eta = 0.1$), we found that coordination has two effects on clumping. First, the fitness cost of clumping is more severe in coordinated filaments than in randomly specialising filaments (Fig. 6d). This occurs because coordinated helpers also invest in signalling molecules and so produce less of the public good than randomly specialised helpers, which amplifies the costs of clumping. Second, coordination leads to a large reduction in the average size of clumps, and so the cost associated with larger clumps is almost never paid (Fig. 6c, d). Consequently, coordination ($s^* > 0$, $v^* > 0$) can produce a substantial fitness advantage in spatial groups by decreasing the chance that costly helper clumps can form and grow.

**Discussion**

Our analyses provide a theoretical framework to help explain why different species of microorganisms use different mechanisms to divide labour[2]. Coordinated division of labour is more likely to be favoured when: (i) social groups are small; (ii) helping is more "essential"; and (iii) there is a low metabolic cost to coordination. While testing our predictions with a formal comparative analysis would require data from more species, our predictions can help to

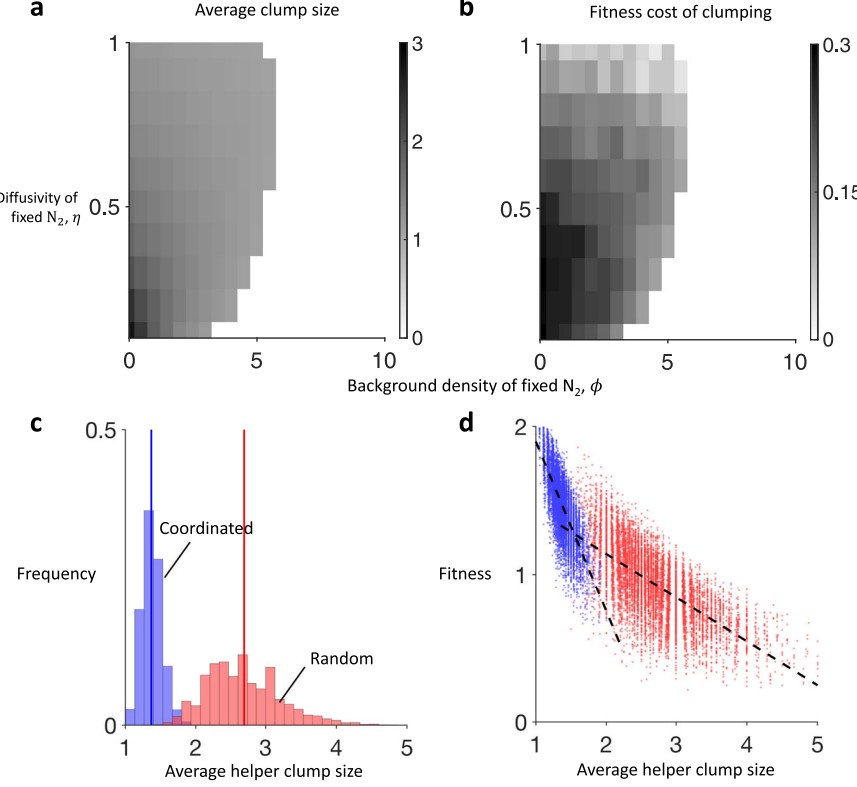

**Fig. 6 Spatial structure and helper clumping.** In randomly specialising filaments, a smaller background density of fixed $N_2$ (smaller $\phi$) and more limited diffusion of helper-fixed $N_2$ (smaller $\eta$), lead to filaments with: **a** a larger average clump size, measured as the average numberx of helpers per clump; and **b** a higher fitness cost of clumping, measured as the slope of the least-squares linear regression of relative fitness on average clump size. We constructed **a**, **b** by performing 1000 independent simulations of growing filaments for each parameter combination, where the trait values are set to the associated optima for random specialisation determined in the previous analysis ($q = q^*, s = 0, d = 0$, and $v = 0$). We then performed 5000 independent simulations of both coordinated (blue) and random (red) filament growth at the extreme case of essential cooperation ($\phi = 0$) and very limited diffusion of fixed $N_2$ ($\eta = 0.1$). **c** Coordination leads to a dramatic reduction in average clump sizes across filaments (average clump size for random (red): 1.4 helpers and coordinated (blue): 2.7 helpers). **d** The absolute fitness cost of larger clumps is greater for coordination specialisation (blue) than for random specialisation (red) but filaments that pay the higher cost of coordination are rare. Slope of least-squares linear regression for random: −0.30 and coordinated: −1.1. Mean squared error of fit for random: 0.033 and coordinated: 0.019. Source data for panels **c** and **d** are provided as a Source Data file. Further details are given in Methods: The effect of helper clumping.

understand the mechanisms that have evolved in well-studied examples.

There are many reasons why coordinated specialisation was favoured to evolve in cyanobacteria filaments. First, cyanobacteria only divide labour when fixed $N_2$ is growth-limiting and so the relative importance of cooperation is high (low $\phi$ and high $h/b$)[9,58,63]. Second, the fixed nitrogen produced by helpers diffuses along the filament, preferentially aiding nearby reproductives and so the effective social group size is small (low $\eta$ and small $n$)[9,49,67]. Third, the initial costs of coordination may have been quite small as new cells could use the local level of fixed $N_2$ as a cue (low $\eta$)[68]. Finally, cyanobacteria filaments have a rigid spatial structure with local benefits from cooperation and thus random specialisation could have led to the accumulation of large sterile clumps, which is a very inefficient distribution of phenotypes (high functional or stochastic cost; Fig. 6).

Colonies of *Volvox carteri* and *Dictyostelium discoideum* use coordination to divide labour, despite the fact that these groups are composed of large numbers of cells (high $n$; on the order of 1000 s of cells or more)[20,69–71]. This highlights that no single factor can fully explain empirical patterns, and that further factors not captured by simple models might be relevant in specific cases. For instance, colonies of *Volvox carteri* require a specific spatial distribution of flagella beaters across the group, which may create a strong selection pressure for coordination, analogous to the avoidance of clumps in

cyanobacteria filaments. Furthermore, in some cases, details of the mechanism of division of labour are still not well understood. For instance, it is possible that there is also an initially random component to pre-stalk specialisation in *Dictyostelium*[70].

There are multiple reasons why random specialisation would have been favoured to evolve in other well-studied species. In *Salmonella enterica*, the self-sacrificing helper cells provide a competitive advantage that eliminates other microbes but is not "essential" to the replication of *Salmonella* cells (lower $h/b$)[12,13]. Further, the benefits of cooperation are provided to all cells in the co-infection ($\eta = 1$) and so the effective social group size is reasonably large (higher $n$). Finally, *Salmonella* pathogens do not have a rigid spatial structure and so there is no scope for the accumulation of growing helper clumps as for cyanobacteria filaments. In *Bacillus subtilis*, a subset of cells become helpers that produce and secrete protein degrading proteases[44]. However, these helper cells are not sterile and so the consequence of deviating from the optimal caste ratios is reduced (Supplementary Methods B.4).

To conclude, most previous work on phenotypic heterogeneity has tended to be either mechanistic, focusing on how different phenotypes are produced (caste determination), or evolutionary, focusing on why heterogeneity is favoured in the first place[1–6,8,11,15,23–28,30,49,70,72–77]. We have used evolutionary models to address the broader question of why different mechanisms are used in different species[2,3,12,23–25]. Focusing on

the reproductive division of labour in microorganisms, we have shown that coordinated specialisation is more likely to be favoured over random specialisation in small groups, when relative coordination costs are low, and when there are larger fitness costs to deviating from optimal caste ratios. We have also shown how these patterns can hold in groups with spatial structure, where there can be a large pressure for an even distribution of phenotypes. These results identify social and environmental factors that could help to explain the distribution of mechanisms to produce phenotypic heterogeneity that has been observed in bacteria, other microbes, and beyond. Aside from microorganisms, our results also suggest a hypothesis for why random caste determination has not been widely observed in animal societies. During the initial evolution of complex animal societies, group sizes were likely to be small and the relative costs of coordination might have been minor compared to each individual's day-to-day organismal metabolic expenditure.

## Methods

### Well-mixed groups

*Labour dividers and their fitness.* We assume that a single individual arrives on an empty patch and, through a fixed series of replications, forms a clonal group of $n$ individuals, $k$ of which are sterile helpers and $n-k$ of which are pure reproductives, where $k \in \mathcal{N} = \{0, 1, 2, \ldots, n\}$. The average fecundity (fitness) of the group in the absence of mechanism costs, $g_{k,n}$, is measured by the per cell number of offspring that would disperse at the end of the life cycle. Denoting by $f_{k,n}$ the fecundity of each reproductive in the group in the absence of fecundity costs, the fecundity of the group is given by Eq. 1. We assume that $f_{k,n}$ depends only on the proportion of helpers in the group, $p = k/n$, with $p \in \mathcal{P}\{0, \frac{1}{n}, , \frac{2}{n}, \ldots, \frac{n-1}{n}, 1\}$, so that we can write

$$f_{k,n} = F(p), \qquad (6)$$

where $F$ is a real function. We further assume that $F$ is increasing on the interval $[0, 1]$ (i.e., the fecundity of each reproductive is increasing in the proportion of helpers in the group). We can then rewrite Eq. 1 as

$$g_{k,n} = (1 - k/n)F(k/n) = G(k/n), \qquad (7)$$

where we have defined

$$G(p) = (1 - p)F(p). \qquad (8)$$

*Fully coordinated specialisation.* With fully coordinated specialisation (C), we assume that some mechanism, such as signalling between cells, ensures that groups always form with the optimal proportion of helpers, $p^*$, where

$$p^* = \frac{k^*}{n}, \text{ with } k^* = \arg\max_{k \in \mathcal{N}} g_{k,n}. \qquad (9)$$

The fitness of a group of coordinated specialisers with an optimal proportion of helpers, $p^*$, is given by Eq. 2, where

$$g_{k^*,n} = G(p^*) \qquad (10)$$

and $0 \le c_C \le 1$ is the metabolic cost of coordination. We make no further assumptions on the functional form of $c_C$. However, we note that it could in principle depend on other model parameters, such as group size $n$.

*Random specialisation.* With random specialisation (R), each individual in the group independently becomes a helper with probability $q$, and a reproductive otherwise. Hence, the number $K$ of helpers in the group is a binomial random variable with parameters $n$ and $q$ (i.e., $K \sim \text{Binomial } (n, q)$). In the following, it will also be convenient to write $Q = K/n$ for the random variable giving the proportion of helpers in the group. Then, the expected fitness of a group of random specialisers is given by

$$w_R(q) = \sum_{k=0}^{n} \binom{n}{k} q^k (1-q)^{n-k} (1-c_R) g_{k,n}, \qquad (11)$$

where $0 \le c_R \le 1$ is the metabolic cost of random specialisation. We assume that this cost is independent of the number of helpers $k$. In contrast to Eq. 3, we have not yet specified here that $q = p^*$.

*Linear public goods.* Our main model assumes that the fecundity function $F$ is given by

$$F(p) = b + hp, \qquad (12)$$

where $b \ge 0$ is a parameter that quantifies the baseline fecundity of reproductives in the absence of cooperation, and $h > 0$ is the scale of the benefits from increased cooperation (higher proportion of helpers). If there is no baseline fecundity

($b = 0$), cooperation by helpers is essential (i.e., the fecundity of reproductives is positive if and only if there are helpers around). If $b > 0$, cooperation is non-essential, with a lower value of $b$ or a higher value of $h$ leading to a larger relative importance of cooperation for the fecundity of reproductives. When cooperation is non-essential ($b > 0$), the ratio $h/b$ is a useful metric for the relative importance of cooperation. Substituting Eq. 12 into Eq. 6 gives Eq. 4.

Replacing (12) into (8) we obtain

$$G(p) = (1 - p)(b + hp) \qquad (13)$$

To find the fitness of coordinated specialisers, we first approximate $p$ by a continuous variable, and calculate the derivative

$$G'(p) = h - b - 2hp.$$

This derivative is decreasing in $p$ (i.e., $G(p)$ is concave), and has a single root given by

$$\hat{p} = \frac{h - b}{2h} = \frac{1}{2}\left(1 - \frac{b}{h}\right). \qquad (14)$$

Such a root lies in the interval $(0, 1)$ if and only if $h > b$. Otherwise, the maximiser of $G(p)$ (and hence the optimal allocation of helpers) is given by $\hat{p} = 0$ (i.e., it is optimal to have no helpers). To avoid this trivial scenario without division of labour, henceforth we assume that $h > b$ holds. Further, to make progress we approximate the optimal allocation of helpers, $p^*$, by $\hat{p}$. The actual optimal value $p^*$ will be a value near $\hat{p}$ but constrained by the permissible group compositions, since $p^* \in P$ (cf. Supplementary Methods A.1, where we relax the assumption that $p^* \approx \hat{p}$). When cooperation is essential ($b = 0$), $\hat{p} = 1/2$. When cooperation is non-essential ($b > 0$), the approximate optimal proportion (Eq. 14) is an increasing function of $h/b$ with $\lim_{b \to 0} \hat{p} = 1/2$. An approximation to the fitness of fully coordinated specialisers can be obtained by substituting Eq. 10 into Eq. 2 and letting $p^* \approx \hat{p}$:

$$w_C(p^*) = (1 - c_C)G(p^*) \approx (1 - c_C)G(\hat{p}). \qquad (15)$$

To find the fitness of random specialisers, we substitute Eq. 13 into Eq. 7, Eq. 7 into Eq. 11, and simplify to obtain

$$\begin{aligned} w_R(q) &= (1 - c_R)\sum_{k=0}^{n}\binom{n}{k}q^k(1-q)^{n-k}\left(1 - \frac{k}{n}\right)\left(b + h\frac{k}{n}\right) \\ &= (1 - c_R)\left(b\mathrm{E}\left[1 - \frac{k}{n}\right] + h\mathrm{E}\left[\left(1 - \frac{k}{n}\right)\frac{k}{n}\right]\right) \\ &= (1 - c_R)\left(b(1-q) + hq(1-q) - h\frac{q(1-q)}{n}\right) \\ &= (1 - c_R)G(q) - h\mathrm{Var}(Q) \end{aligned} \qquad (16) (17)$$

where we have made use of the first two moments of the binomial distribution, $\mathrm{E}[K] = nq$, $\mathrm{E}[K^2] = nq(1-q) + (nq)^2$ and of the fact that $\mathrm{Var}(Q) = q(1-q)/n$.

In order to determine the condition under which coordinated specialisation is favoured over random specialisation, we assume in a first step that random specialisers play the strategy, $q^*$, so that their fitness is given by

$$w_R(p^*) = (1 - c_R)(G(p^*) - h\mathrm{Var}(P^*)), \qquad (18)$$

where $P^* = K^*/n$ and $K^* \sim \text{Binomial}(n, p^*)$ (and hence $\mathrm{Var}(P^*) = p^*(1 - p^*)/n$). This assumption simplifies our calculations and leads to results that are qualitatively similar to those that arise from the more parsimonious assumption that random specialisers play the strategy that maximises their fitness (cf. Supplementary Methods A.2, where we assume that random specialisers play optimally).

We can evaluate the condition for coordinated specialisation to be favoured over random specialisation (i.e., when $w_C(p^*) > w_R(p^*)$ holds) by comparing Eq. 15 and Eq. 18. The condition is given by

$$\mathrm{Var}(P^*)\frac{h}{G(p^*)} > \frac{c_C - c_R}{1 - c_R} \equiv \gamma. \qquad (19)$$

The left-hand side of this inequality is the normalised fecundity benefit of switching from random specialisation to coordinated specialisation, and the right-hand side of the inequality ($\gamma$) is the normalised relative change in metabolic costs paid from doing so. Inequality 19 shows that the fecundity benefit of coordination over random specialisation can be decomposed into a measure of the deviation from the optimal allocation of labour, $\mathrm{Var}(P^*)$, and a quantity that captures the relative cost of deviating from the optimal proportion of helpers, $h/G(p^*)$.

To obtain a simple expression of Condition 19 in terms of our parameters ($n$, $b$, and $h$) we approximate $p^*$ by $\hat{p}$ as given in Eq. 15 to obtain

$$\mathrm{Var}(P^*) \approx \frac{(h - b)(h + b)}{4nh^2}, \qquad (20)$$

$$\frac{h}{G(p^*)} \approx \frac{4h^2}{(h + b)^2}. \qquad (21)$$

With these approximations, Condition 19 becomes Condition 5. Note that the left-hand side of Condition 5 is increasing in the benefits of cooperation $h$ and decreasing in group size $n$ and the baseline fecundity $b$. Since $G(p^*)$ is

(approximately) independent of $n$, we can say that the effect of increasing the group size acts primarily on the deviation of groups from the optimal proportion of helpers, $\mathrm{Var}(P^*)$. In contrast, a smaller baseline fecundity (lower $b$) or more benefits from cooperation (larger $h$) both (i) push $p^*$ closer to $1/2$ (which in turns increases the variance $\mathrm{Var}(P^*)$) and (ii) increases the cost of deviation (larger $h/G(p^*)$) and thus acts via both factors.

## Cyanobacteria filaments

*Life cycle.* Here, we give details on the processes that govern filament growth in our model. A filament begins as an array of four connected cells and increases in number until it reaches a maximum of $L$ cells, at which point all reproductive cells produce a large number of offspring that disperse to found filaments in the next generation of the group life cycle. The remaining cells then die (i.e., generations are non-overlapping).

We let $L_t$ be the number of cells in the filament at time $t \geq 0$, and $I_t = \{1, \ldots, L_t\}$ be the set of (indexes to) individuals in the filament. Each individual cell has the fixed phenotype of a helper or a reproductive. Further, let $H_t \subset I_t$ and $R_t \subset I_t$ be, respectively, the set of helpers and reproductives in the filament at time $t$. We assume that the two interior cells are reproductives and the two exterior cells are helpers at the start of the filament growth ($t = 0$). That is, we have $I_0 = \{1, 2, 3, 4\}$, $H_0 = \{1, 4\}$, and $R_0 = \{2, 3\}$.

*The local density of the public good across the filament.* The rate at which a reproductive absorbs the public good depends on the local density of the public good at its location in the filament. We assume that the density of the public good at location $i \in R_t$ at time $t$ is equal to

$$\Phi_i^t = \phi + \sum_{j \in H_t} \phi_{i,j}^t, \tag{22}$$

where $\phi \geq 0$ is the uniform background density of the public good due to the environment, and $\phi_{i,j}^t \geq 0$ is the increase in the local density of the public good that is due to helper $j$ at time $t$, which is assumed to be given by

$$\phi_{i,j}^t = \bar{\phi}(1-s)^\zeta \frac{\eta^{|i-j|}}{\sum_{k \in I_t} \eta^{|k-j|}}, \tag{23}$$

where $\bar{\phi}$ is the maximum rate of public good production by a helper, $(1-s)^\zeta$ is the degree to which this production decreases due to a trade-off with the production of signalling molecules (as described below), and the last factor ensures that $\phi_{i,j}^t$ declines by a factor of $0 < \eta \leq 1$ for every cell position that separates the reproductive cell $i$ from the helper $j$ (diffusion factor of fixed nitrogen). When $\eta$ is small, helpers only provide substantial public good benefits to their nearest neighbours. When $\eta$ is large, even reproductives at a considerable distance along the filament receive public good benefits. The denominator of this last factor enforces a conservation principle such that an increase in diffusivity, $\eta$, does not artificially increase the amount of the public good produced by helpers. In the limit as $\eta \to 1$, all reproductives benefit equally from the efforts of each helper.

*Size of reproductives.* Reproductive cells grow over time as they absorb the public good from the environment. We could alternatively conceptualise this process as an increase in energy or resource reserve over time. Let $\pi_i^t$ be the size of reproductive cell $i \in R_t$ at time $t$. We assume that each reproductive starts with base size $\pi_i^0 = 0$. For any time interval $\Delta t$ during which no cell divides anywhere in the filament, the increase in size of a reproductive cell $i \in R_t$ is calculated as

$$\pi_i^{t+\Delta t} = \pi_i^t + \Psi_i^t \Delta t,$$

where $\Psi_i^t$ is the instantaneous growth rate of reproductive $i \in R_t$. We assume that $\Psi_i^t$ is an increasing but diminishing function of the rate of public good that is absorbed at its location, $\Phi_i^t$, according to the functional form

$$\Psi_i^t = \psi\left(1 - e^{-\mu \Phi_i^t}\right), \tag{24}$$

where $\psi$ is the maximum growth rate, and larger $\mu$ leads to a more diminishing curve.

*Replication of reproductives.* Reproductives grow until they reach a critical size $\bar{\pi}$, at which point they divide by budding off a daughter cell to one side of the parent cell along the filament. At any time, $t$, we calculate the time until the next replication, $\tau^t$, using the following procedure. For each reproductive cell $i \in R_t$, we calculate its expected time until replication as

$$\tau_i^t = \frac{\bar{\pi} - \pi_i^t}{\Psi_i^t}, \tag{25}$$

i.e., the amount it has left to grow divided by its growth rate. Here, we have held fixed the growth/replication of all other reproductives. Thus, the next reproductive to divide, in this case, is simply the reproductive with the smallest expected time to replication, $\tau^t = \min_{i \in R_t} \tau_i^t$. When a reproductive cell $i \in R_t$ divides, it buds off a daughter cell either before or after the parent cell along the filament with equal probability. The positions, phenotypes, and sizes of all other cells in the filament are reindexed to account for the new cell (all cells to the right of the daughter cell

move one space further along the indexing array). Whether the daughter cell becomes a helper or a reproductive depends on the mechanism of specialisation (see below). The parent reproductive cell size is reset to zero and, if the daughter is reproductive, its size is set to zero as well. The group life cycle ends when the filament has reached a size of $L$ cells, at which point all reproductives produce a large number of offspring that disperse to found filaments in the next generation of the group life cycle. The remaining cells then die (i.e., generations are non-overlapping).

*How cells specialise.* When a new cell is produced, whether the cell becomes a helper or a reproductive depends on the mechanism of specialisation. There are four co-evolving traits in our model that combined determine the mechanism of specialisation: (i) the baseline probability, $0 \leq q \leq 1$, (ii) the level of signalling, $0 \leq s \leq 1$, (iii) the response sensitivity, $v \geq 0$, and (iv) the response threshold, $d \geq 0$. The baseline probability, $q$, is the probability that the new cell adopts a helper role in the absence of coordination (i.e., if either $s = 0$ or $v = 0$). If $s > 0$, helper cells produce signalling molecules that diffuse along the length of the filament. Let $i$ be the position of the new cell at time $t$ in the filament and where the indices of all other cells have been updated to account for the new cell. The level of the signal detected by the cell is

$$\chi_i^t = \max\left(\lambda s \sum_{j \in H_t} \xi^{|i-j|} + \varepsilon, 0\right), \tag{26}$$

where $\lambda s$ is the maximum rate of signal produced by each helper, and $\xi$ is the factor by which the signal declines for each position that separates the helpers from the receiver. The term $\varepsilon$ is a normal random variable with mean 0 and variance $\sigma_\varepsilon^2$ that accounts for the fact that new cells do not perfectly detect the level of the signal.

The degree to which the new cell responds to the signal when specialising depends on the interaction between the detected level of the signal, $\chi_i^t$, and the trait values $q$, $d$, and $v$. Specifically, we model the probability, $p$, that the cell adopts a helper phenotype via the response norm

$$p = \min\left(1, \max\left(0, q + 1 - \frac{2}{1 + e^{-v(\chi_i^t - d)}}\right)\right). \tag{27}$$

The form of this function is depicted in Fig. 4b. We assume that if helpers emit no signal, then the response threshold $d$ is also held at zero (i.e., $d = 0$ if $s = 0$). Random specialisation occurs if cells are insensitive to the signal (i.e., $v = 0$) or if cells send no signal at all (i.e., $s = 0$). In this case, Eq. 27 becomes $p = q$, and new cells adopt a helper phenotype with the baseline probability $q$. Coordination occurs when $v > 0$ and $s > 0$, in which case the probability of adopting a helper phenotype is affected by the detected level of the signal, $\chi_i^t$. The larger the response sensitivity, $v$, and the larger the difference between the response threshold, $d$, and the detected level of the signal, $\chi_i^t$, the more the probability of adopting a helper phenotype $p$ is perturbed from the baseline probability $q$. If the detected signal level is greater than the response threshold (i.e., if $\chi_i^t$), then sensitive cells (with $v > 0$) decrease their probability of adopting a helper phenotype (and thus $p < q$). If the detected signal level is less than the response threshold (i.e., if $\chi_i^t < d$), then sensitive cells (with $v > 0$) increase their probability of adopting a helper phenotype (and thus $p > q$). In the limit, as cells become infinitely sensitive ($v \to \infty$), the response norm leads to a deterministic response where if the detected signal level is smaller than the threshold, the new cell always becomes a helper ($p = 1$), and if the detected signal level is greater than the threshold, the new cell always becomes a reproductive ($p = 0$).

In our model, higher coordination incurs larger metabolic costs. First, the more helpers invest in producing the signal (higher $s$) the fewer resources they have to produce the public good (Eq. 25). Second, we assume that filaments with more sensitive cells (higher $v$) grow more slowly, which we model as an increase in the growth-cap of reproductives, via

$$\bar{\pi} = \bar{\pi}_0 + e^{\beta v} - 1, \tag{28}$$

where $\bar{\pi}_0$ is the baseline growth-cap. More sensitive cells (higher $v$) lead to an exponentially increasing growth-cap with shape parameter $\beta$.

Consequently, in different scenarios, the optimal trait values $q$, $s$, $d$ and $v$, will depend on the trade-offs as the filament grows between producing more helpers and producing more reproductives, and between the growth costs and the advantages of coordination.

*Evolution of coordination.* For a given set of model parameters, we estimate the evolutionarily optimal trait values $q^*$, $s^*$, $d^*$ and $v^*$ by simulating an evolving population of cyanobacteria filaments over 4000 generations. We initialise the population with the resident trait values all equal to zero ($q = s = d = v = 0$). Each generation, we consider an invading mutant strategy, which we draw from a multivariable random distribution, $\mathrm{Normal}(\boldsymbol{\mu}, \boldsymbol{\Sigma})$, with location $\boldsymbol{\mu}$ equal to a vector

containing the resident trait values $(q, s, d, v)^\top$ and with a covariance matrix

$$\boldsymbol{\Sigma} = \begin{pmatrix} \sigma_q^2 & 0 & 0 & 0 \\ 0 & \sigma_s^2 & 0 & 0 \\ 0 & 0 & \sigma_d^2 & 0 \\ 0 & 0 & 0 & \sigma_v^2 \end{pmatrix}. \tag{29}$$

We do not allow all traits to mutate at the same time (but see Supplementary Fig. 6). For the first 500 generations, we consider only mutations in the baseline helper probability $q$, by setting $\sigma_s^2 = \sigma_d^2 = \sigma_v^2 = 0$. This allows the population to evolve to the optimal strategy for random specialisation. For the rest of the generations, we allow only the coordination traits $s$, $d$, and $v$ to mutate by removing the constraint on $\sigma_s^2, \sigma_d^2$ and $\sigma_v^2$ and by instead constraining $\sigma_q = 0$. We set the resident trait value of $q$ to be its average trait value over generations 250 to 500. In addition, we set $d = 0$ whenever $s = 0$ to avoid neutral drift of $d$.

For each generation of the simulation, we estimate the fitness of the resident strategy and the mutant strategy by simulating the growth of 200 independent filaments using each strategy and averaging fitness across simulations. For a given simulation, let $\tau_L$ be the time at which the $L$th cell in the filament is produced. We calculate fitness as

$$w_{\text{filament}} = \frac{\sum_{i \in R_{\tau_L}} \Psi_i^{\tau_L}}{\tau_L}, \tag{30}$$

that is, as the sum of the fecundities of the reproductives in the last generation of the group life cycle (estimated as their growth rates), divided by the time it took the filament to grow to that size. If the estimated fitness of the mutant strategy is greater than the estimated fitness of the resident strategy, we replace the resident strategy with the mutant strategy before proceeding to the next generation. If the estimated fitness of the mutant strategy is less than the estimated fitness of the resident strategy, we keep the resident strategy when proceeding to the next generation. We approximate the evolved trait values $(q^*, s^*, d^*,$ and $v^*)$ for each evolutionary simulation as the average of each trait value over the last 2000 generations of the simulation.

*Simulation results.* There are 15 parameters in our model (see Supplementary Table). We focused our investigation on the particular patterns produced by two of these parameters, namely $\phi$ and $\eta$. The other parameters and simulation parameters were fixed to values given in the Supplementary Table. In Supplementary Fig. 5, we provide some sample plots to illustrate the evolutionary convergence of our simulations for the specific case of essential cooperation ($\phi = 0$) and very local cooperation ($\eta = 0.1$). We see that all trait values converge to the approximate final rolling average within 100–200 generations of being allowed to evolve. The results for the evolved level of signalling, $s^*$, response sensitivity, $v^*$, baseline helper probability, $q^*$, and response threshold, $d^*$ are shown in Fig. 5. In these results, the evolved trait values $(q^*, s^*, d^*$ and $v^*)$ are averages across 5 independent evolutionary simulations for each parameter combination.

For each parameter combination examined, we performed further simulations to estimate the extent that the evolved level of coordination lead to a more or less precise division of labour. For each case, we ran $T = 10,000$ independent simulations of cyanobacteria growth using the evolved strategies $(q^*, s^*, d^*$ and $v^*$; Fig. 5a–d). For each of the $T$ simulated filaments, indexed by $i \in \{1, \dots, T\}$, we recorded both the total number of helpers in the last generation of the filament, $H_i$, and the number of helpers in the leftmost non-terminal 10 cells in the last generation of the filament, $\widetilde{H}_i$ (excluding the outside helper). We calculated the relative variance in the number of helpers in the leftmost non-terminal 10 cells as

$$\text{Relative variance} = \frac{\text{Var}_i(\widetilde{H}_i)}{10\bar{h}(1 - \bar{h})}, \tag{31}$$

where the average proportion of helpers in the last generation across all $T$ simulations is given by $\bar{h} = \frac{1}{T}\sum_i \frac{H_i}{L}$. The numerator of Eq. 31 is the observed variance in the number of helpers in the leftmost non-terminal 10 cells. The denominator of Eq. 31 is the variance of a binomial distribution with 10 trials and probability of success equal to $\bar{h}$, which is what we would expect if cells specialise randomly and independently from one another. If the level of coordination produces an allocation of labour that is indistinguishable from random specialisation, then the relative variance should be approximately one. As the level of coordination produces more and more precise allocations of labour, the relative variance is expected to decline. The precision of coordination as shown in Fig. 5 and Supplementary Fig. 6 corresponds to the reciprocal of the relative variance.

We also considered the possibility that filaments begin with no helpers. We repeated the above analyses, while ignoring the parameter combinations in which cooperation is essential ($\phi = 0$). Results for the evolved trait values and relative precision are given in Supplementary Fig. 6 and show similar qualitative patterns.

*The effect of helper clumping.* We ran additional simulations to quantify the propensity and cost of helper clumps.

For a given set of parameter values (Supplementary Table) we extracted the evolved trait values from the previous set of simulations and ran $T$ independent simulations of filament growth with the associated evolved strategy $(q^*, s^*, d^*$

and $v^*$). When examining random specialisation, we set $s^* = d^* = v^* = 0$. For each simulation, we define a clump as any contiguous grouping of helpers in the last generation of the group growth. Thus, clump sizes can range from 1 (a single helper) to $L$ (the entire filament). Within each individual simulation, indexed as $i \in \{1, \dots, T\}$, we calculated the average clump size, $m_i$, over all clumps in the filament. The average clump size across all $T$ simulations is then the cross-simulation average, $\bar{m} = \frac{1}{T}\sum_i m_i$. To determine the cost of clumping, we also recorded the fitness of each filament, $w_i$, for all $T$ simulations (Equation 33). The cost of clumping is then approximated as the slope of the linear least-squares regression of filament fitness on average clump size, given by

$$\text{Cost of clumping} = \frac{\text{Cov}(w_i, m_i)}{\text{Var}_i(m_i)}, \tag{32}$$

In Fig. 6a, b, we used the above approach to map the propensity and cost of clumping from random specialisation as a function of the background density of fixed nitrogen ($\phi$) and the diffusivity of fixed nitrogen ($\eta$). In Fig. 6c, d, we used the above approach to determine the differences between random specialisation and coordinated specialisation, focusing on the extreme case of essential cooperation $\phi = 0$ and very low diffusivity of fixed nitrogen ($\eta = 0.1$).

We then determined the effect of group growth on the formation of helper clumps in randomly specialising filaments. To do this, we ran $T$ independent simulations where the filament does not grow and is composed of $L$ cells. An individual cell becomes a helper with a probability equal to the optimal strategy for the growing group, $q^*$, and otherwise, it becomes a reproductive. We then quantified the average clump size in each simulation in the same way as for the previous analysis. Supplementary Fig. 7 shows the distribution of average clump sizes across $T = 10,000$ simulations of non-growing filaments (Supplementary Fig. 7A) and growing filaments (Supplementary Fig. 7B). This was calculated for the extreme case of essential cooperation $\phi = 0$ and very low diffusivity of fixed nitrogen ($\eta = 0.1$). We find that non-growing groups still form helper clumps but that the distribution has a smaller average value and has a smaller upper tail than for growing filaments. Consequently, helper clumping is more severe when cellular division and differentiation are "coupled".

**Reporting summary.** Further information on research design is available in the Nature Research Reporting Summary linked to this article.

## Data availability

Source data for Fig. 6 and Supplementary Fig. 7 are provided as a Source Data file. Further data generated in this study are available on Github (https://doi.org/10.5281/zenodo.5747159). Source data are provided with this paper.

## Code availability

All simulated data were generated using C and Matlab. The codes used for this study are available on Github (https://doi.org/10.5281/zenodo.5747159).

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

## Acknowledgements

We thank: Anna Dewar, Kevin Foster, Andy Gardner, Ashleigh Griffin, Asher Leeks, Matishalin Patel, Tom Scott, and Daniel Unterwegger for their helpful comments and suggestions; St. John's College, Oxford (GAC), the French National Research Agency (ANR) under the Investments for the Future (Investissements d'Avenir) program (grant ANR-17-EURE-0010 to the IAST; J.P.) and the ERC (Horizon 2020 Advanced Grant 834164; S.A.W.) for funding. We would like to acknowledge the use of the University of Oxford Advanced Research Computing (ARC) facility in carrying out this work. (https://doi.org/10.5281/zenodo.22558).

## Author contributions

G.A.C., S.A.W., and J.P. conceived the study. G.A.C. and J.P. designed and analysed the analytical models, G.A.C. and M.L. designed and analysed the simulation models. G.A.C. and S.A.W. wrote the first draft. All authors contributed toward writing the final manuscript.

## Competing interests

The authors declare no competing interests.
