## [Peer Review File · Nature Communications]

Reviewers' Comments:

Reviewer #1:

Remarks to the Author:

In this manuscript, the authors present a simple mathematical model with the aim to delimit conditions under which evolution would favor different mechanisms to determine the different phenotypes involved in division of labor. The authors focus on the case of reproductive division of labor in microbes that occur in clonal colonies.

This is an interesting question and the paper is well written. The main conclusions are:

- 1) Smaller costs of coordination favor coordinated specialization.
- 2) Smaller social groups favor coordinated specialization.
- 3) More essential public goods favor coordinated specialization.

Given how the model is constructed all three predictions are intuitive (but see next point). In particular, since the model assumes that coordinated specialization is intrinsically superior to random specialization prediction 1 has to be true almost by definition.

I am a bit unsure about the implementation of "essentiality" and to what extent it drives the third prediction. Eq. (4) implies that the fecundity of reproductives is a monotonically decreasing function of epsilon, i.e., the essentiality of cooperation. I guess this assumption does not affect prediction 3 qualitatively but it should affect it quantitatively. Is this effect of epsilon on reproduction desired? Why does it make sense? I wonder whether this assumption could introduce a bias toward coordination with increasing epsilon that has nothing to do with the respective advantages and disadvantages of coordinated and random specialization?

The model builds on a crucial assumption that is discussed very little. The authors assume that random specialization comes for free and only coordinated specialization incurs a cost. This is assumed to be the case even though the probability for an individual to adopt one of the two phenotypes equals the optimal frequency of phenotypes. Thus, what is assumed here is that each cell has a genetically encoded "weighted coin" where the weights correspond to the optimal frequency of the two phenotypes. This raises two questions. First, do cells have an easy way to "build" such a weighted coin? Second, is building such a coin cost-free? Neither of these questions is discussed by the authors. Answering them in the negative should have strong implications for the construction of a model.

This brings me to a related question. Would it be in some way possible to quantify the relative importance of the factors 1-3? I imagine that features of the life cycle of an organism can have a strong effect of which model of phenotype determination is actually evolutionarily attainable, an aspect that is largely captured by the cost parameter. As a consequence, I imagine that in most examples mentioned in table 1 it is the feasibility of the alternative modes of phenotype determination that was the main driver in its evolution, while group size might play a much smaller role. Discussing this would be important.

The simulation model builds on the assumption that cells interact pairwise and that any two cells in a group have the same probability to interact with each other. I believe that this assumption is both highly specific and rather unrealistic for most colonial microbes. First, this would require a well mixed population of cells, which is certainly not the case in biofilms or colonies with other fixed structures (spherical as in *Volvox* or filamentous as in *Anabaena*). Second, even in a well mixed population I am unsure that the exchange of information is accurately approximated as a pairwise interaction. I understand that this is a way to formulate a simple model but I would argue that it is not in agreement with the idea that the model is particularly general. Rather, I think what the authors hope is that this is a good approximation for a wider range of possible interactions but this is unclear.

From the description of the model in the main text several important aspects of the simulation model are completely unclear. For example, it is unclear why a low value of s corresponds to random specialization since the text says that the decision about specialization occurs after an interaction event. Second, it is completely unclear how evolution is implemented in this model.

These things are explained in the appendix but I believe that the main part of a paper should be self-contained to a higher degree than is the case here. I assume this brevity is due to constraints from the journal but this is certainly disadvantageous for readability.

Some patterns emerging from comparing Figure 4D and Figure S5 are puzzling/interesting. First, there seems to be parameter combinations where $s^* > 0$ but $t^* = 0$. My understanding is that this combination amounts to producing no helper cells but with paying some cost for communication. It seems that this combination does not make sense biologically. Is the explanation that what appears as $t^* = 0$ in figure S5 is in fact $0 < t^* < 0.1$? Second, overlaying Figures 4D and S5 allows to delimit (at least) four regions: (i) perfect coordinated specialization, (ii) mixture between coordinated and random specialization, (iii) pure random phenotype determination, and (iv) absence of helpers. Would it not be useful to highlight all of these regions in the graphical representation of the simulation results?

I understand that the purpose of table 1 is to check whether a broad qualitative agreement exists between the predictions of the model and what is known from the biology of some model systems. According to the authors (II. 22-23 and II. 380-383) such an agreement indeed exists.

I am wondering whether the authors interpret the data somewhat in their favor and whether it would not equally be possible to come to the opposite conclusion. For instance, three of the four species with coordinated specialization are classified as consisting of large groups, counter to the prediction of the model. Furthermore, it seems that it would be easily possible to come to somewhat different classifications with respect to some entries in the table.

1)

Wikipedia tells me that colonies of *Volvox carteri* consist of 2000-6000 cells while fruiting bodies of *Myxococcus xanthus* consist of up to 100 000 cells, yet bodies species are classified as having a large group size.

2)

In II. 478-480 *B. subtilis* and *A. cylindrical* are treated in the same sentence with the same conclusion, yet, in one species the essentiality of cooperation is classified as Medium and in the other as Medium/High.

3)

In I. 487 it is mentioned that both *S. enterica* and *B. subtilis* are free-floating but my understanding is that both can occur in biofilms. In this case, according to the logic presented in Appendix H the costs could be Medium?

I also note that the range of group sizes considered in the model analysis (1-40 individuals) is likely several orders of magnitude lower than group sizes in microbes.

MINOR POINTS

I. 24 and II. 82-85: "how evolutionary theory can help explain both why between-individual coordination evolves"

I am not sure what the authors precisely want to say here. The paper presents a simple model to answer a clear question, just as many other papers do. The "broader question" raised in II. 82-85 is in my opinion non-existent. Why should evolutionary theory not be suitable to explain the question raised in this manuscript, No theoretician would doubt that "evolutionary theory" in the form of a mathematical model could be developed for the question posed in this manuscript.: "how evolutionary theory can help explain both why between-individual coordination evolves"

and

II. 82-85

I am not sure what the authors precisely want to say here. The paper presents a simple model to answer a clear question, just as hundreds of other papers do. The "broader question" raised in II. 82-85 is in my opinion nonexistent. Why should evolutionary theory not be suitable to explain the question raised in this manuscript? No theoretician would doubt that "evolutionary theory" in the form of a mathematical model could be developed for the question posed in this manuscript.

Is the division of labor occurring in *Bacillus subtilis* and described in II. 177-179 really a reproductive division of labor? This is not clear from the text here.

Eq. (5). What does it actually mean in this model when the right-hand side of inequality (5) is negative? In other words, what should I expect for $\epsilon < 1/2$? Is it that all cells are reproductives, i.e., $k=0$?

II. 269-271

Here the authors seem to say that simpler models are more general. But is this really true? I wonder whether the simple model presented here is in a certain sense not also highly specific but it is just that this specificity can be described by simple equations. I would argue that a general model would be a model that leaves functional forms unspecified and just makes some assumptions about properties of these functions (like the sign of its derivatives). What I do agree with is that this model is not built to match any particular system.

I. 285

that it would be useful -> that would be useful

This sentence is in fact slightly misleading. I don't think the authors would like to investigate the assumptions theoretically but rather the implications of these assumptions.

Reviewer #2:

Remarks to the Author:

Review report Nature communications

In their manuscript "Mechanisms to divide labour in social microorganisms" the authors present a model that studies the mechanisms underlying the division of labor in microbes. The authors specifically focus on two mechanisms, random specialization and coordinated specialization, and ask what type of specialization is favored by selection. To derive general insights, the authors purposely formulate a generic model that sidesteps the regulatory particularities of any known microbial system. They assume cells form clonal groups with a constant size. Within these groups, cells play a public goods game that favors the presence of non-reproducing helper cells. Using slightly different implementations of their model, both analytical and numerical, the authors show that coordinated specialization evolves when (1) group sizes are small, (2) cooperation is essential and (3) costs of coordination are low. The authors conclude their analysis by comparing their three modeling predictions with empirical data and claim that their results are roughly supported by empirical examples of microbial division of labor.

I applaud the research question addressed by the authors, which I think is both timely and relevant. The mechanisms of specialization are at the core of our understanding of how division of labor evolves in microbes. In addition, the authors do a great job in communicating their model assumptions and results. Yet, unfortunately, I also think the model has some major shortcomings that demand considerable revision.

Major comment 1. While the title of the manuscript gives the impression that the authors are specifically concerned with the division of labor in microbes, their modeling assumptions are only weakly motivated by microbial systems. I understand there is value in formulating a generic model on the division of labor, but I think the model assumptions should (at least partly) be backed up by empirical examples, which is now arguably insufficiently the case. For example:

(1) The authors consider groups up to 40 cells, which is extraordinarily small with regard to most microbial systems studied empirically: *Salmonella enterica* colonies, *Myxococcus xanthus* fruiting bodies, *Bacillus subtilis* colonies, *Volvox carterii* individuals all consist of hundreds if not thousands of individual cells. With these large group sizes, according to the model of the authors, random specialization should outperform coordinated specialization, how do the authors explain the occurrence of coordinated specialization in larger groups? Does their model provide additional insight when it comes to such larger groups?

(2) The authors only consider a reproductive division of labor. Although this form of division of labor is observed in eusocial insects and many advanced forms of multicellularity (e.g. plants and animals), it is rare among microbes. Even among the examples listed in Table 1, not all species

have a strict reproductive division of labor. Why did the authors choose to study this form of division of labor? How would their modeling results change when helper cells can also reproduce?

(3) The authors assume that the group size is constant. Although I understand that this simplifies the analysis, it implies that the different types of specialization do not affect the growth dynamics within a patch, which is unlikely to be the case. For example, if – due to random specialization – a cell that colonizes a patch becomes a helper cell, it would strongly impair growth and decrease patch fitness. Why did the authors not consider implementing more realistic growth dynamics in their model? What would happen if the model accounts for within-patch growth?

(4) The authors assume that the costs of coordinated specialization increase monotonically with the group size, with nearly no costs for small groups and high costs for large groups. It is unclear how this cost function is motivated. For example, in many cases, microbial coordination comes about through quorum-sensing. The costs of quorum-sensing do not increase with group size. In other words, the per capita costs of coordination are expected to be the same whether the group consists of ten cells or hundreds of cells. If the costs of coordination do not scale with group size, would coordination still be favored in small groups only?

(5) In their simulations, the authors also consider partial coordination, where the level of coordination, s , varies from being entirely random ($s=0$) to being fully coordinated ($s=1$). s is equal to the probability that two cells interact with each other. If a cell has no interaction partners within the group, it specializes randomly, if it has interaction partners, it specializes deterministically depending on the fraction of helper cells. To my knowledge, this implementation of partial coordination does not correspond to any real-world example. That is, there are no microbial system, where a fraction of cells specializes randomly and another fraction deterministically within the same group. I would argue that a reaction norm model would be more appropriate here, where the probability of specialization can depend on the fraction of helper cells within the group. This could lead to random specialization (no dependency), partially coordinated specialization (weak dependency) or fully coordinated specialization (strong dependency). Such implementation would more closely reflect the regulatory systems studied empirically and still provide general insights.

Major comment 2. The authors conclude their manuscript by comparing their modeling predictions with empirical findings. The authors provide “rough qualitative (verbal) estimates” of the group size, essentiality of cooperation, costs of coordination and type of specialization, and present their results in Table 1. Yet, I think their classification highly problematic and fails in nearly every category.

The results in Table 1 contradict the conclusions the authors draw in the main text. That is, the authors conclude that “Consistent with our predictions, coordination appears to be more prevalent in species where group sizes are smaller, cooperation is more essential, and/or the cost of coordination is lower”. While in Table 1, only one of the four species that is claimed to have coordinated specialization is classified as having a small group size (*A. cylindrica*). Why do the authors claim that their modeling predictions are consistent with the table?

Then regarding the different categories:

(1) Group size. The group sizes of species are much higher than those considered by the model (with the exception of *Anabaena cylindrica*), but the authors do not discuss this. The authors claim that *Bacillus subtilis* has a medium group size, because the effective radius of cooperation is determined by the diffusion coefficient of protein degrading proteases. Why do the authors assume that these proteins only diffuse locally? Has this ever been quantified? To my knowledge, small proteins should rapidly diffuse at the timescale of colony growth. Also, why do the authors even consider *B. subtilis*, as it does not have a reproductive division of labor?

(2) Essentiality of cooperation. This category is perhaps most speculative. Whether or not cooperation is essential ultimately depends on the ecology in which specialization evolved. For most of the species in Table 1, we have surprisingly little knowledge about their ecology. Without this knowledge, how can we even determine if fruiting body formation in *Myxococcus* and *Dictyostelium* is more essential than cell differentiation in *Salmonella*? I think the current

classification hinges on tautological reasoning: species that rely on coordinated specialization are argued to more strongly depend on cooperation, but this could well be the consequence of evolving the coordinated specialization as opposed to being the cause.

(3) Costs of coordination. Also for this category, the authors wildly speculate. The authors claim that the costs of coordination depend on the distance between cells and the metabolic complexity of cells. Why these criteria are considered most relevant remains unclear. For example, regarding the distance, why is cell-cell communication in dense microbial colonies less costly than in liquid culture? The authors also state that "S. enterica and B. subtilis are free-floating", which is incorrect. Both species are potent colony formers, and Salmonella is known to form aggregates during infection. Finally, the authors seem to assume that communication forms a major cost for coordination, but also this is not supported by the literature they present.

(4) Types of specialization. The classification between random specialization and coordinated specialization might be misleading to the layman reader. For example, even though Salmonella and Bacillus are classified under random specialization, cell differentiation in these species is still subject to regulation. In fact, depending on the environmental conditions, the fraction of specialized cell types can dramatically change. Although the signaling cascades are only partly understood, classifying them as 'random specialization' may give the false impression to the reader that there is no regulatory control. Instead of including Table 1, I would suggest the authors to provide a rigorous review of the existing literature, so that the reader can appreciate how random specialization and coordination come about in well-studied microbial systems.

Overall, I think the classification in Table 1 is far too speculative for publication. I would therefore recommend removing this section altogether and instead attempt on improving the empirical motivation of the model.

Minor comments.

-Figure legends of Figure 4B and C contain mistake.

Reviewer Comments:

Reviewer #1 (Remarks to the Author):

In this manuscript, the authors present a simple mathematical model with the aim to delimit conditions under which evolution would favor different mechanisms to determine the different phenotypes involved in division of labor. The authors focus on the case of reproductive division of labor in microbes that occur in clonal colonies.

This is an interesting question and the paper is well written.

We thank the reviewer for this kind assessment.

The main conclusions are:

- 1) Smaller costs of coordination favor coordinated specialization.
- 2) Smaller social groups favor coordinated specialization.
- 3) More essential public goods favor coordinated specialization.

Given how the model is constructed all three predictions are intuitive (but see next point). In particular, since the model assumes that coordinated specialization is intrinsically superior to random specialization prediction 1 has to be true almost by definition.

We have included a sentence to more clearly link prediction 1 to our model's assumptions (Lines 218-220).

I am a bit unsure about the implementation of "essentiality" and to what extent it drives the third prediction. Eq. (4) implies that the fecundity of reproductives is a monotonically decreasing function of epsilon, i.e., the essentiality of cooperation. I guess this assumption does not affect prediction 3 qualitatively but it should affect it quantitatively. Is this effect of epsilon on reproduction desired? Why does it make sense? I wonder whether this assumption could introduces a bias toward

coordination with increasing epsilon that has nothing to do with the respective advantages and disadvantages of coordinated and random specialization?

We have addressed this concern by reformulating the model, replacing the essentiality parameter with two separate parameters, which we hope are more intuitive: the baseline fecundity of reproductives, b , and the scale of the benefits from cooperation, h . (Lines 175-181).

When $b=0$, cooperation is essential, and when $b>0$ cooperation is non-essential and the ratio h/b provides a useful metric for how essential cooperation is. Thus, a decreasing baseline fecundity b leads both to a monotonically decreasing fecundity of reproductives while also making cooperation more “essential”. The previous model is the specific case where we set $b=1-h$;

The model builds on a crucial assumption that is discussed very little. The authors assume that random specialization comes for free and only coordinated specialization incurs a cost. This is assumed to be the case even though the probability for an individual to adopt one of the two phenotypes equals the optimal frequency of phenotypes. Thus, what is assumed here is that each cell has a genetically encoded “weighted coin” where the weights correspond to the optimal frequency of the two phenotypes. This raises two questions. First, do cells have an easy way to “build” such a weighted coin? Second, is building such a coin cost-free? Neither of these questions is discussed by the authors. Answering them in the negative should have strong implications for the construction of a model.

We have reformulated the analytical model to address this concern by explicitly including both a metabolic cost of coordination, c , and a metabolic cost of random specialisation, d , (Lines 144-145 and Line 155). This improves the analysis as it allows us to highlight how the relative difference in metabolic costs affects which mechanism is favoured. For instance, random specialisation could only ever be favoured if the metabolic costs of random specialisation are less than the metabolic costs of

liberated specialisation (Lines 207-228). We surmise that this is a

We posit as given in this work that cells that specialise randomly can have a genetically encoded “weighted coin” to do so. We outline in the text that this involves amplification of intra-cellular noise with a genetic positive feedback circuit (Lines 40-45 and 94-97). More specific mechanistic details of how this works are given in the literature that we cite (references 7,8, 29-33).

This brings me to a related question. Would it be in some way possible to quantify the relative importance of the factors 1-3? I imagine that features of the life cycle of an organism can have a strong effect of which model of phenotype determination is actually evolutionarily attainable, an aspect that is largely captured by the cost parameter. As a consequence, I imagine that in most examples mentioned in table 1 it is the feasibility of the alternative modes of phenotype determination that was the

main driver in its evolution, while group size might play a much smaller role. Discussing this would be important.

This is a good point, we thank the reviewer for raising it.

In the discussion of prediction 1, we now highlight that random specialisation can never evolve unless the metabolic cost of coordination is greater than the metabolic cost of random specialisation ($>$) (Lines 212-220). This provides a necessary condition for random specialisation that must be satisfied before the other predictions can have an effect. In the empirical discussion we suggest that this may be the case for division of labour in *D. discoideum* and *V. carteri* (Lines 468-478).

The simulation model builds on the assumption that cells interact pairwise and that any two cells in a group have the same probability to interact with each other. I believe that this assumption is both highly specific and rather unrealistic for most colonial microbes. First, this would require a well mixed population of cells, which is certainly not the case in biofilms or colonies with other fixed structures (spherical as in *Volvox* or filamentous as in *Anabaena*). Second, even in a well mixed population I am unsure that the exchange of information is accurately approximated as a pairwise interaction. I understand that this is a way to formulate a simple model but I would argue that it is not in agreement with the idea that the model is particularly general. Rather, I think what the authors hope is that this is a good approximation for a wider range of possible interactions but this is unclear.

We have developed a new more mechanistic model for the secondary analysis that allows for fixed within-group spatial structure, based explicitly on the biology of filamentous cyanobacteria. This is both more spatially specific and more realistic. In addition, this model includes details for how coordination can occur through the use of signalling molecules that diffuse along length of the filament.

As the reviewer suggests, the previous simulation was intended as a good, albeit rough, approximation of a wider range of possible systems. Consequently, we highlight this model's results in the main text (lines 302-305) but moved its details to supplementary section F.

From the description of the model in the main text several important aspect of the simulation model are completely unclear. For example, it is unclear why a low value of s corresponds to random specialization since the text says that the decision about specialization occurs after an interaction event.

A low value of s in the previous model captures the probability that a cell is connected to another cell and thus bases its specialisation decision on the phenotype of other cells. A low s values implies a higher chance of independent, random specialisation at the individual level.

Second, it is completely unclear how evolution is implemented in this model. These things are explained in the appendix but I believe that the main part of a paper should be self-contained to a higher degree than is the case here. I assume this

brevity is due to constraints from the journal but this is certainly disadvantageous for readability.

We have developed a new model for the secondary analysis and include more details of how evolution is implemented for this model in the main text (Lines 383-394).

Some patterns emerging from comparing Figure 4D and Figure S5 are puzzling/interesting. First, there seems to be parameter combinations where $s^* > 0$ but $t^* = 0$. My understanding is that this combinations amounts to producing no helper cells but with paying some cost for communication. It seems that this combination does not make sense biologically. Is the explanation that what appears as $t^* = 0$ in figure S5 is in fact $0 < t^* < 0.1$?

Yes, we believe that this is an artifact of how we “bucket” the heat maps, and that these are parameter combinations that lead to a very low number of helpers.

Second, overlaying Figures 4D and S5 allows to delimit (at least) four regions: (i) perfect coordinated specialization, (ii) mixture between coordinated and random specialization, (iii) pure random phenotype determination, and (iv) absence of helpers. Would it not be useful to highlight all of these regions in the graphical representation of the simulation results?

We thank the reviewer for the suggestion. We have labelled the broad strategy outcomes in the results figure for the new simulation model (Figure 5C).

I understand that the purpose of table 1 is to check whether a broad qualitative agreement exists between the predictions of the model and what is know from the biology of some model systems. According to the authors (ll. 22-23 and ll. 380-383) such an agreement indeed exists.

I am wondering whether the authors interpret the data somewhat in their favor and whether it would not equally be possible to come to the opposite conclusion. For instance, three of the four species with coordinated specialization are classified as consisting of large groups, counter to the prediction of the model. Furthermore, it seems that it would be easily possible to come to somewhat different classifications with respect to some entries in the table.

1)

Wikipedia tells me that colonies of *Volvox carteri* consist of 2000-6000 cells while fruiting bodies of *Myxococcus xanthus* consist of up to 100 000 cells, yet bodies species are classified as having a large group size.

2)

In ll. 478-480 *B. subtilis* and *A. cylindrical* are treated in the same sentence with the same conclusion, yet, in one species the essentiality of cooperation is classified as Medium and in the other as Medium/High.

3)

In l. 487 it is mentioned that both *S. enterica* and *B. subtilis* are free-floating but my understanding is that both can occur in biofilms. In this case, according the logic presented in Appendix H the costs could be Medium?

The kind of comparative analysis we would suggest to test our predictions must often rely on rough approximations, where finer details between species are averaged out in the analysis, and can often still lead to interesting results.

However, our attempt to qualitatively categorise our parameter values in different microbial species was heuristic and led to criticisms from both reviewers. We think that their comments are a fair and so we have removed Table 1 from the manuscript and instead include a discussion of how our predictions can help to better understand key examples of either mechanism, while emphasising that much more quantitative data is needed before a formal test of our predictions is feasible.

I also note that the range of group sizes considered in the model analysis (1-40 individuals) is likely several orders of magnitude lower than group sizes in microbes.

We have expanded the group sizes considered in the presentation of our results (group sizes up to 1000 cells now shown in Figure 3 and in many of the supplementary figures, particularly those with analytical solutions). We note that our qualitative results hold for even larger group sizes but that a smaller relative cost of coordination would be required in order to see an interesting contrast between optimal mechanisms at this range.

MINOR POINTS

I. 24 and II. 82-85: “how evolutionary theory can help explain both why between-individual coordination evolves”

I am not sure what the authors precisely want to say here. The paper presents a simple model to answer a clear question, just as many other papers do. The “broader question” raised in II. 82-85 is in my opinion non-existent. Why should evolutionary theory not be suitable to explain the question raised in this manuscript, No theoretician would doubt that “evolutionary theory” in the form of a mathematical model could be developed for the question posed in this manuscript

We have changed the language to be more clear on the “adaptationist” argument we are trying to make (Line 23-25).

Is the division of labor occurring in *Bacillus subtilis* and described in II. 177-179 really a reproductive division of labor? This is not clear from the text here.

Yes, we have altered the language to make this more clear (Line 170-173). Reference 40 in particular has more details on this.

Eq. (5). What does it actually mean in this model when the right-hand side of inequality (5) is negative? In other words, what should I expect for $\epsilon < 1/2$? Is it that all cells are reproductives, i.e., $k=0$?

Yes, ($h < b$) in our reformulated model means that there are no helpers ($k=0$). We have included an additional sentence to make this more clear (Line 188-191).

II. 269-271

Here the authors seems to say that simpler models are more general. But is this really true? I wonder whether the simple model presented here is in a certain sense not also highly specific but it is just that this specificity can be described by simple equations. I would argue that a general model would be a model that leaves functional forms unspecified and just makes some assumptions about properties of these functions (like the sign of its derivatives). What I do agree with is that this model is not built to match any particular system.

Our updated manuscript contains both 1) a simple analytical model that is not built to match any particular system, but which provides useful predictions with factors that are easy to interpret and apply across different systems, and; 2) a more detailed simulation model that is explicitly based on the biology of growing cyanobacteria filaments, and is particular to that system. The combination of both analyses is intended to show that the predictions of our model can apply both in specific cases and be useful in explaining diversity more generally.

I. 285

that it would be useful -> that would be useful

This sentence is in fact slightly misleading. I don't think the authors would like to investigate the assumptions theoretically but rather the implications of these assumptions.

Yes, agreed. This paragraph has been removed to reduce the length of the manuscript.

Reviewer #2 (Remarks to the Author):

In their manuscript "Mechanisms to divide labour in social microorganisms" the authors present a model that studies the mechanisms underlying the division of labor in microbes. The authors specifically focus on two mechanisms, random specialization and coordinated specialization, and ask what type of specialization is favored by selection. To derive general insights, the authors purposely formulate a generic model that sidesteps the regulatory particularities of any known microbial system. They assume cells form clonal groups with a constant size. Within these groups, cells play a public goods game that favors the presence of non-reproducing helper cells. Using slightly different implementations of their model, both analytical and numerical, the authors show that coordinated specialization evolves when (1) group sizes are small, (2) cooperation is essential and (3) costs of coordination are low. The authors conclude their analysis by comparing their three modeling predictions

with empirical data and claim that their results are roughly supported by empirical examples of microbial division of labor.

I applaud the research question addressed by the authors, which I think is both timely and relevant. The mechanisms of specialization are at the core of our understanding of how division of labor evolves in microbes. In addition, the authors do a great job in communicating their model assumptions and results.

We thank the reviewer for this kind assessment.

Yet, unfortunately, I also think the model has some major shortcomings that demand considerable revision.

We hope that the changes to the analysis and the new model address these concerns.

Major comment 1. While the title of the manuscript gives the impression that the authors are specifically concerned with the division of labor in microbes, their modeling assumptions are only weakly motivated by microbial systems. I understand there is value in a formulating generic model on the division of labor, but I think the model assumptions should (at least partly) be backed up by empirical examples, which is now arguably insufficiently the case. For example:

(1) The authors consider groups up to 40 cells, which is extraordinarily small with regard to most microbial systems studied empirically: *Salmonella enterica* colonies, *Myxococcus xanthus* fruiting bodies, *Bacillus subtilis* colonies, *Volvox carteri* individuals all consist of hundreds if not thousands of individual cells. With these large group sizes, according to the model of the authors, random specialization should outperform coordinated specialization, how do the authors explain the occurrence of coordinated specialization in larger groups? Does their model provide additional insight when it comes to such larger groups?

We have extended our analysis to present results for larger group sizes (up to 1000 cell groups, Figure 3, Supplementary Figures 1 and 3). In some cases, when simulations or numerical calculations are required our analysis is constrained to smaller group sizes (Supplementary Figures 2 and 3) but we expect that our patterns should hold as group size is scaled up, particularly if we assume small coordination costs. For the analytical results, our predictions will hold qualitatively for any group size (Condition 5).

In the empirical discussion, we offer an explanation for why some species with large group sizes employ coordinated specialisation (468-478).

(2) The authors only consider a reproductive division of labor. Although this form of division of labor is observed in eusocial insects and many advanced forms of multicellularity (e.g. plants and animals), it is rare among microbes. Even among the examples listed in Table 1, not all species have a strict reproductive division of labor. Why did the authors choose to study this form of division of labor?

We may have a difference in terminology with the reviewer here. In our work, reproductive division of labour occurs whenever a subset of the group specialises more on costly cooperation (to whatever degree) whereas the other subset focuses more on reproduction. Critically, the helper cells need not be sterile but can simply have a reduced fecundity relative to the purely reproductive cells. The presence of sterile helpers is simply an extreme form of reproductive division of labour.

There are number of important examples of reproductive division of labour in microbes across bacteria, simple forms of multicellular algae and unicellular eukaryotes. For instance, all of the examples in our previous table 1 as well further examples in a previous review (reference 2). Our model focuses in particular on sterile helpers to simplify the analysis and because there are many instances of reproductive division of labour in microbes with sterile or suicidal helpers.

How would their modeling results change when helper cells can also reproduce?

This is a good question. We have developed an additional model to examine the impact of non-sterile helpers (Lines 289 and 294-300, supplementary section E.4).

We find qualitatively the same results, while also revealing that more specialised helpers (closer to sterility) are more likely to favour coordination. We also found an interesting result that a higher relative importance of cooperation (h/b) can sometimes disfavour coordination, particularly when helpers are quite fecund. This is contrary to the prediction for sterile helpers, but also makes quite a bit of sense. We discuss this additional result in the main text (Lines 294-300) and further details are given in supplementary section E.4. and supplementary figure S3C and S3D.

(3) The authors assume that the group size is constant. Although I understand that this simplifies the analysis, it implies that the different types of specialization do not affect the growth dynamics within a patch, which is unlikely to be the case. For example, if – due to random specialization – a cell that colonizes a patch becomes a helper cell, it would strongly impair growth and decrease patch fitness. Why did the authors not consider implementing more realistic growth dynamics in their model? What would happen if the model accounts for within-patch growth?

These are good questions. To answer them, we have: (1) developed an additional analysis (Lines 289-290, supplementary section E.5) that investigates the impact of growth dynamics in the simple model; and (2) our new simulation model based on cyanobacteria filaments includes the effects of group-growth dynamics (Lines 307-448; supplementary section G).

Broadly, we find similar results but also that within group growth with otherwise rigid spatial structure and local cooperation is particularly unfavourable for random specialisation (Figure 5, Lines 439-448).

(4) The authors assume that the costs of coordinated specialization increase monotonically with the group size, with nearly no costs for small groups and high costs for large groups. It is unclear how this cost function is motivated. For example, in many cases, microbial coordination comes about through quorum-sensing. The costs of quorum-sensing do not increase with group size. In other words, the per capita costs of coordination are expected to be the same whether the group consists of ten cells or hundreds of cells. If the costs of coordination do not scale with group size, would coordination still be favored in small groups only?

We have reformulated the model so as to remove any specific assumptions about the shape of the metabolic cost of coordination, μ , and how it could depend on other parameter models (Lines 144-145). This shows that our predictions for group size do not rely on an interaction with the cost of coordination.

Empirically, how coordination costs might increase with larger group sizes depend on the details of the biological system in question and how coordination is achieved. For instance, if diffusing signalling molecules are used and all cells must receive them for coordination to be successful, then a larger group size requires a significantly larger investment in the signal. However, if coordination only requires signalling between nearest neighbours in the group, then no such increase in investment would necessarily be expected.

(5) In their simulations, the authors also consider partial coordination, where the level of coordination, s , varies from being entirely random ($s=0$) to being fully coordinated ($s=1$). s is equal to the probability that two cells interact with each other. If a cell has no interaction partners within the group, it specializes randomly, if it has interaction partners, it specializes deterministically depending on the fraction of helper cells. To my knowledge, this implementation of partial coordination does not correspond to any real-world example. That is, there are no microbial system, where a fraction of cells specializes randomly and another fraction deterministically within the same group. I would argue that a reaction norm model would be more appropriate here, where the probability of specialization can depend on the fraction of helper cells within the group. This could lead to random specialization (no dependency), partially coordinated specialization (weak dependency) or fully coordinated specialization (strong dependency). Such implementation would more closely reflect the regulatory systems studied empirically and still provide general insights.

We have replaced the secondary analysis in the main text with a more specific model of division of labour, based on the biology of cyanobacteria filaments (Lines 307-448). This model is more explicit about the mechanism of coordination. As the reviewer suggests, we employed a reaction norm curve to capture how newly formed cells respond the local density of signalling molecules and how different levels of sensitivity (dependency) to the signal could arise (Figure 4B). We thank the reviewer for this suggestion.

Major comment 2. The authors conclude their manuscript by comparing their modeling predictions with empirical findings. The authors provide “rough qualitative (verbal) estimates” of the group size, essentiality of cooperation, costs of coordination and type of specialization, and present their results in Table 1. Yet, I think their classification highly problematic and fails in nearly every category.

The results in Table 1 contradict the conclusions the authors draw in the main text. That is, the authors conclude that “Consistent with our predictions, coordination appears to be more prevalent in species where group sizes are smaller, cooperation is more essential, and/or the cost of coordination is lower”. While in Table 1, only one of the four species that is claimed to have coordinated specialization is classified as

having a small group size (*A. cylindrica*). Why do the authors claim that their modeling predictions are consistent with the table?

As we discussed in the above response, Table 1 was intended to provide a rough sense for how the model parameters might be distributed across labour dividing microbe species, while acknowledging that it is not a formal test of our predictions. We agree that our classifications are both heuristic and frequently debatable. We have removed Table 1 from the manuscript so as to avoid any possible confusion and so as not to make any controversial/debatable empirical claims.

Then regarding the different categories:

(1) Group size. The group sizes of species are much higher than those considered by the model (with the exception of *Anabaena cylindrica*), but the authors do not discuss this. The authors claim that *Bacillus subtilis* has a medium group size, because the effective radius of cooperation is determined by the diffusion coefficient of protein degrading proteases. Why do the authors assume that these proteins only diffuse locally? Has this ever been quantified? To my knowledge, small proteins should rapidly diffuse at the timescale of colony growth.

Table 1 removed. Some research has found that public goods concentrations decrease rapidly as you move away from the producer (see for instance reference 60). This particular finding seems to arise primarily due to the uptake of the good by other cells. More geometrically, if the good is diffusing in a 3D environment, then density of the good would naturally decline with distance. Volume (the denominator of density in this case) increases by a power of three as distance from the producer increases.

Also, why do the authors even consider *B. subtilis*, as it does not have a reproductive division of labor?

In *B. subtilis* populations there is a reproductive division of labour as only a subset of non-sporulating cells produce and secrete protein degrading enzymes, which benefit the local population of cells (reference 40).

(2) Essentiality of cooperation. This category is perhaps most speculative. Whether or not cooperation is essential ultimately depends on the ecology in which specialization evolved. For most of the species in Table 1, we have surprisingly little knowledge about their ecology. Without this knowledge, how can we even determine if fruiting body formation in *Myxococcus* and *Dictyostelium* is more essential than cell differentiation in *Salmonella*?

Table 1 removed. We assumed here that the benefits of avoiding starvation are larger than the benefits of out-replicating competitors.

I think the current classification hinges on tautological reasoning: species that rely on coordinated specialization are argued to more strongly depend on cooperation, but this could well be the consequence of evolving the coordinated specialization as opposed to being the cause.

This is a fair point, and tricky to disentangle. To avoid this issue we have removed Table 1.

(3) Costs of coordination. Also for this category, the authors wildly speculate. The authors claim that the costs of coordination depend on the distance between cells and the metabolic complexity of cells. Why these criteria are considered most relevant remains unclear. For example, regarding the distance, why is cell-cell communication in dense microbial colonies less costly than in liquid culture? The authors also state that “*S. enterica* and *B. subtilis* are free-floating”, which is incorrect. Both species are potent colony formers, and *Salmonella* is known to form aggregates during infection. Finally, the authors seem to assume that communication forms a major cost for coordination, but also this is not supported by the literature they present.

Table 1 removed. When cells are not physically attached or sharing a common developmental programme, the cost of communication is the metabolic cost of producing and secreting signalling molecules, which must be more numerous the farther the signal must travel and the more cells that it must reach.

(4) Types of specialization. The classification between random specialization and coordinated specialization might be misleading to the layman reader. For example, even though *Salmonella* and *Bacillus* are classified under random specialization, cell differentiation in these species is still subject to regulation. In fact, depending on the environmental conditions, the fraction of specialized cell types can dramatically change. Although the signaling cascades are only partly understood, classifying them as ‘random specialization’ may give the false impression to the reader that there is no regulatory control.

This is an important point of potential confusion and we thank the reviewer for pointing it out. Coordination and regulation are not the same thing. The probability that a random specialiser adopts a helper role can still be regulated, so long as it is regulated by an environmental cue and not in response to a signal produced by another cell (in which case that would be coordination). We include a sentence to this effect in lines 160-163.

Instead of including Table 1, I would suggest the authors to provide a rigorous review of the existing literature, so that the reader can appreciate how random specialization and coordination come about in well-studied microbial systems.

We have removed Table 1, as described above.

Overall, I think the classification in Table 1 is far too speculative for publication. I would therefore recommend removing this section altogether and instead attempt on improving the empirical motivation of the model.

We have replaced Table 1 with an empirical discussion of key examples (Lines 450-490). We have expanded and clarified our definitions of coordinated and random specialisation at the start of the manuscript (Lines 33-36 and Lines 40-

42), while also retaining our key empirical examples of each (Lines 36-40 and Lines 42-45) as well as the illustrative Figures 1 and 2 and their legends.

However, to limit the manuscript length we have decided not to add a more thorough review of the existing literature. We cite many primary works and more in-depth reviews that provide further details on how the mechanistic details work to produce specialisation in different individual species (For instance, refs 1-11, 16-21, 23-24, 29-33, ...).

Minor comments.

-Figure legends of Figure 4B and C contain mistake.

Thank you for pointing this out. The legend has been fixed in what is now Figure S4.

Reviewers' Comments:

Reviewer #1:

Remarks to the Author:

This is the revision of a manuscript I reviewed previously. The authors investigate different processes that can determine alternative phenotypes in colonial organisms. The authors focus on microorganism, specifically bacteria and single celled eukaryotes, that form groups of genetically identical cells in one way or the other. The authors seem to propose that in such organisms a certain cell dimorphism is common: cells develop into either reproductive cells or helper cell. Specifically, the authors seem to propose that it is common that helper cells produce a common good that is used by reproductive cells and in exchange give up (or reduce) their ability to reproduce themselves. In the language of multicellular organisms, this corresponds to germ line and soma cells. In other words, the premise of this manuscript is that helper cells have evolved to increase the success of reproductive cells and that this a common phenomenon in micro-organisms. Given this premise, the manuscript investigates under what conditions either random or coordinated phenotype determination is selectively favoured. Specifically, the authors propose that the key factors in determining the evolution of different phenotype determining mechanisms are (i) the costs of having a sub-optimal ratio of reproductive and helper cells, (ii) the physiological costs for random and coordinated phenotype determination, and (iii) the importance of cooperation for group fitness.

To me it seems an open question how widespread division of labour in microbes between reproductive cells and a common good producing helper cells really is. The authors list five examples that became model systems due their division of labor it is unclear to what extent these examples are representative for a larger class of microbes.

I am furthermore skeptical that the general model (Eqs. 1-5), which aims to explain evolved mechanisms of phenotype determination as functions of the above mentioned three factors, captures the key selective forces at work in the examples discussed by the authors. I will details my concerns in the following.

Volvox

Colonies of *Volvox carteri* consist of a multi-cellular spheric outer hull of helper cells that in their interior carry few germ cells. This is clearly an example for a coordinated phenotype. But I am sceptical that a key-driver in this body design is to produce the optimal ratio of helper and reproductive cells or the physiological costs of random and coordinated phenotype determination. Rather, a random assemblage of reproductive and helper cells would result in a non-functional phenotype. I would therefore suggest that the key driver for coordinated phenotype determination in this example it to produce a functional bauplan.

Bacillus subtilis

Reference (40) from this paper reports the finding that *B. subtilis* cells express protease at varying levels and the authors of that paper raise the possibility that protease producing cells act altruistically. In that study cells are cultured in a liquid medium in which they seem to flow more or less freely. If this is the case, then it seems to me that the lifecycle excludes coordinated production of protease cells. Furthermore, I could not find any indication that cells differ in their level of reproduction and it is therefore unclear to me whether it is justified to classify *B. subtilis* as species with reproductive division of labor.

Salmonella enterica

Division of labour occurs between self-sacrificing cells that enter the gut tissue and reproducing cells that stay in the gut lumen. To me, it seems likely that the random phenotype determination that is present in this example can again most easily explained by their function. The two different cell types move independently, self-sacrificing cells enter the gut tissue while reproducing cells stay in the gut lumen, and I cannot envisage how these different activities could be usefully combined with coordinated phenotype determination. Here, a functional bauplan is likely more easily achieved with random phenotype determination than with coordinated phenotype determination.

Filamentous cyanobacteria

This case is possibly the most convincing example of a species in which the coordinated determination of phenotypes is partly driven to ensure an optimal ratio of nitrogen to non-nitrogen fixing cells. But in agreement with the authors, I would argue that in this example the optimal spatial arrangement of the two cell types is of even greater importance. This is captured in the new individual-based model presented in the manuscript. However, another aspect of the biology of filamentous cyanobacteria is strangely lacking from this model. My understanding of the literature is that the main explanation for the evolution of two cell types in cyanobacteria is the fact photosynthesis and nitrogen fixation are two physiologically incompatible tasks. Thus, the general view seems to be that division of labour is between nitrogen fixing and photosynthesising cells, not between nitrogen fixing and reproductive cells. Thus, I presume that there is bi-directional exchange of products, photosynthesising cells are also delivering their products to the nitrogen fixing cells. This process is ignored by the authors in their detailed model. If their aim is to make a system specific model, I believe that it has to be included since this ought to be an additional factor avoiding the clumping of nitrogen fixing cells.

Dictyostelium discoideum

Division of labor in this example occurs between non-viable stalk cells that hold up and help disperse viable spore cells. This example is similar to Volvox in the sense that I believe producing the optimal ratio between reproductive and helper cells is likely not important for the evolution coordinated cell type determination. Rather, deterministic cell type determination is essential to produce a functional phenotype (stalk and fruiting body).

In summary, the authors argue for the importance of group size, cost of random and coordinated phenotype determination and importance of cooperation for the evolution of different mechanisms for phenotype determination. However, I believe that the examples given in the manuscript might be better explained by functional constraints.

I like the simplicity of their simple model but it is rather unclear to me whether it has any explanatory power for the standard microbial examples of division of labour. At this stage, I wonder whether system specific models as the one suggested for filamentous cyanobacteria are a more fruitful avenue (although such a model might have to contain additional details).

Additional comments on the detailed model for filamentous cyanobacteria.

From reading the main text it did not become clear to me how is it determined which exact reproductive cell undergoes the next cell division? Is this completely random or does it depend on its internal level of N_2 ?

II. 387-389

Does a biological motivation exist for this simulation algorithm or is it chosen just for convenience?

Does the optimal proportion q^* depend on η , the cooperation range?

It is not obvious to me that a smaller η is analogous to a smaller social group (smaller n ; line 419). If the optimal q^* is independent of η , then q^* is determined by the total colony ontogeny and maybe the maximal group size. In particular, the sentence "With random specialisation, a smaller social group can lead to proportions of helpers that show greater deviations from the optimum, increasing the benefit that can be obtained by coordination (Figure 3)" would need further explanations.

The effect of ϕ does not seem to be explored in the simulations, and the remark in II. 434-437 seems therefore a bit unconnected.

Minor points

Throughout the manuscript the use of italics for mathematical symbols is applied inconsequential.

The equation at the end of I. 143 is displayed incompletely but I think it is given in Eq. S5 in the Appendix.

I. 159

"sometimes" should be "most of the time". The probability that exactly np^* helpers are produced goes to zero as n goes to infinity.

I. 193: It is not obvious to me why the expression $(h-b)/n(h+b)$ represents the fecundity benefits of switching from random specialisation to coordination.

I. 220 sometimes -> rarely

Reviewer #2:

Remarks to the Author:

I am pleased to see that authors substantially revised their manuscript on two important fronts. First, the authors provide a more nuanced evaluation of the empirical literature in relation to their model, which is less speculative and does better justice to the scope of their model. Second, besides a number of extensions on their phenomenological model, the authors now also included a more mechanistic model that examines how a division of labor can evolve in Cyanobacteria. This model recapitulates some of the more trivial predictions of the phenomenological model (e.g. coordination is more likely to evolve at low costs and high relative benefits of cooperation) and also reveals some differences, which makes it an interesting addition. Most importantly, the mechanistic model shows that coordination could evolve not only because it leads to the correct fraction of specialized cells, but also because it could lead to an even distribution of those cells. With random specialization, the same fraction of specialized cells leads to reduced benefits of cooperation, because those specialized cells tend to clump more, which effectively reduces the associated benefits. This mechanistic model shows that one should be cautious with generalizing insights obtained from phenomenological models.

Although I am pleased with the revised manuscript, I unfortunately think an additional revision is required. My comments pertain mostly to the newly introduced model on Cyanobacteria that comes with a number of ad hoc assumptions that are insufficiently well motivated and may even require revision.

(1) The authors assume that signal production reduces nitrogen fixation of the helper cells, while signal sensitivity delays timing of replication (modeled through an increased growth cap at division). It is unclear why increased sensitivity would change the timing of division. Would it not be more likely that the costs of sensitivity, which may be caused by increased expression of receptor proteins, result in a reduced growth rate, rather than changing the cell size at division. Why did the authors make this assumption and how are their results different when sensitivity affects growth rate directly?

(2) Why do the authors assume that hormogonia give rise to filaments with specialized cells at the tips of the filaments. What happens when authors start from undifferentiated filaments, as observed in Cyanobacteria? Since coordinated spacing requires the presence of specialized cells, I can imagine that starting from undifferentiated filaments necessitates a higher rate of stochastic differentiation.

(3) As already mentioned above, I think that it is interesting that coordination evolves for a wider range of parameter conditions in the mechanistic model than in the phenomenological model, because there is an additional benefit of an even distribution of specialized cells. Could authors quantify the costs of clumping? Do I understand correctly that clumping is particularly common in the model because cell differentiation is coupled to division? Would you still see pervasive clumping when cell differentiation is uncoupled from division?

(4) It is unclear why authors first let the differentiation probability (q) evolve for 500 generations and then the remaining parameters underlying coordination. Why do these parameters not evolve simultaneously? If q can evolve simultaneously with the other parameters, would the parameter space for which coordination evolves become larger or smaller?

(5) I would suggest the authors to show all four evolving parameters in Figure 5?

(6) The authors say "Colonies of *Volvox carteri* and *Dictyostelium discoideum* use coordination to divide labour, despite the fact that these groups are composed of large numbers of cells (high n ; on the order of 1000s of cells or more). However, the initial specialisation of helpers in each case appears to make use of a shared developmental programme and so the relative costs of coordination could be quite small for both species (low γ)"

I do not understand how authors can draw this conclusion. How is the developmental programme shared by cells in *Dictyostelium* or *Volvox* different from that shared by cells within a filament in Cyanobacteria? I consider it way more plausible that the phenomenological model, as presented by the authors, simply fails to capture all the benefits that could drive the evolution of coordination. In the phenomenological model, coordination provides a single benefit only, by leading to the correct fraction of specialized cells. However, as the authors also show with their model on Cyanobacteria, other benefits might follow from coordination – such as an even distribution of specialized cells. The phenomenological model might simply fail to capture such benefits, leaving it unexplained why coordination could also evolve in larger groups. I would strongly encourage the authors to better highlight the limitations of their phenomenological model.

(7) The authors continue by saying that "Further, in both cases an absence of helpers would lead to cell death and thus cooperation is important (higher h/b)"

As pointed out previously, here again the authors risk tautological reasoning: specialization evolved, because specialization is essential, but specialization might as well have become essential after it first evolved (from a non-essential state). In fact, the latter is much more plausible for *Dictyostelium*, as the ancestor of *Dictyostelium* was hypothesized to be capable of surviving starvation through encystation (which is widely conserved among the solitary and many social slime molds), from which fruiting body formation and sporulation later evolved. By the same token, in some species of Cyanobacteria, knocking out heterocyst differentiation would lead to cell death in the absence of a nitrogen source, but in many other species, there are no specialized cells, because vegetative cells simply cycle between photosynthesis and nitrogen fixation.

REVIEWER COMMENTS

Reviewer #1 (Remarks to the Author):

This is the revision of a manuscript I reviewed previously. The authors investigate different processes that can determine alternative phenotypes in colonial organisms. The authors focus on microorganism, specifically bacteria and single celled eukaryotes, that form groups of genetically identical cells in one way or the other. The authors seem to propose that in such organisms a certain cell dimorphism is common: cells develop into either reproductive cells or helper cell. Specifically, the authors seem to propose that it is common that helper cells produce a common good that is used by reproductive cells and in exchange give up (or reduce) their ability to reproduce themselves. In the language of multicellular organisms, this corresponds to germ line and soma cells. In other words, the premise of this manuscript is that helper cells have evolved to increase the success of reproductive cells and that this a common phenomenon in micro-organisms. Given this premise, the manuscript investigates under what conditions either random or coordinated phenotype determination is selectively favoured. Specifically, the authors propose that the key factors in determining the evolution of different phenotype determining mechanisms are (i) the costs of having a sub-optimal ratio of reproductive and helper cells, (ii) the physiological costs for random and coordinated phenotype determination, and (iii) the importance of cooperation for group fitness.

To me it seems an open question how widespread division of labour in microbes between reproductive cells and a common good producing helper cells really is. The authors list five examples that became model systems due their division of labor it is unclear to what extend these examples are representative for a larger class of microbes.

The referee appears to be questioning how often reproductive division of labour has been observed in microbes. While we only discussed a few canonical examples in detail, there are many more.

This is an exciting and rapidly progressing area of research, where new examples are being discovered frequently. However, even 5-6 years ago there were sufficient examples for broad reviews (Ackerman, 2015, *Nature Reviews Microbiology* 13, 497-508; West & Cooper *Nature Reviews Microbiology* 14, 716-723). The papers which cite these reviews are filled with examples of more recent examples. For instance, an exciting very recent study is division of labour in antibiotic production in *Streptomyces* (Zhang et al., 2020, *Science Advances*, 6, eaay5781). The lesson seems to be that microorganisms engage in complex social behaviours to a much larger extent than we would have thought possible even a few years ago.

Irrespective of exactly how common division of labour turns out to be, there are clear examples in different species with different mechanisms, and this variation needs explaining. In addition, theory is often used to develop hypotheses *a priori*, that can then be tested with further data collection. Furthermore,

microorganism groups that engage in reproductive division of labour are of fundamental interest because they can help shed light on the ancestors of multicellular organisms.

I am furthermore skeptical that the general model (Eqs. 1-5), which aims to explain evolved mechanisms of phenotype determination as functions of the above mentioned three factors, captures the key selective forces at work in the examples discussed by the authors. I will details my concerns in the following.

In the following part of their review, the referee goes through some specific worries about several empirical examples. There are two general problems that repeatedly undermine these concerns:

First, with no empirical basis, the referee is disagreeing with the conclusions made in the original empirical papers. For example, the referee claims a lack of evidence for division of labour in *Bacillus subtilis*, despite a substantial experimental literature (e.g. Kovacs et al. 2018 *Nature Microbiology* 3, 1451-1460; Dragos et al. 2018 *Current Biology* 28, 1-11; van Gestel et al. 2015 *PLoS Biology* 13, e1002141; Chai et al. 2007 *Molecular Microbiology* 67, 254-263; Lopez et al. 2008, *FEMS microbiology reviews* 33, 152-163; Veening et al. 2008, *Molecular systems biology*, 4(1), 184).

Second, the referee appears to be arguing that instead of using evolutionary theory to examine the cost and benefits of different mechanisms (strategies), that the mechanism used by each species can instead be explained as selection for a 'functional bauplan' as opposed to a 'non-functional phenotype'.

We disagree with this suggestion, which we think is based on a fundamental misunderstanding of evolutionary theory. In particular, the referee's suggestion is not a competing hypothesis, it is just using different words to describe the same thing. The referee is still suggesting that natural selection will favour traits that lead to higher fitness (functional bauplan) over traits that lead to lower fitness (non-functional phenotype). The referee seems to be suggesting that huge fitness differences are required (functional versus non-functional) – whereas natural selection will favour traits even if they have small fitness benefits. Our model captures both large and small fitness benefits, and so can cover all possibilities.

A related issue is that the referee leans on unsubstantiated verbal arguments. The benefit of theoretical modelling, as opposed to a verbal argument, is that it forces the author to make all assumptions explicit. This purpose of evolutionary theory, relative to verbal arguments, has been explained and reviewed numerous times, since the 1970s (e.g. Maynard Smith 1982 *Evolution and the Theory of Games*).

The referee emphasises that multiple factors can influence fitness. We agree completely – the aim of our modelling is to examine the impact of these multiple factors, and how they interact.

Volvox

Colonies of *Volvox carteri* consist of a multi-cellular spheric outer hull of helper cells that in their interior carry few germ cells. This is clearly an example for a coordinated phenotype. But I am sceptical that a key-driver in this body design is to produce the optimal ratio of helper and reproductive cells or the physiological costs of random and coordinated phenotype determination. Rather, a random assemblage of reproductive and helper cells would result in a non-functional phenotype. I would therefore suggest that the key driver for coordinated phenotype determination in this example is to produce a functional bauplan.

See above. In addition, we agree that the main model does not account for further costs associated with where in the group the helpers are located. However, capturing the impact of such spatial structure was part of the motivation for developing the simulation model (Line 314-315). While this is based on a simpler system (filaments), we found and emphasised that the need for helpers to be located in the correct locations does strongly favour coordination (Lines 540-543). Consequently, our analyses account for all of the reviewer's objections here.

Bacillus subtilis

Reference (40) from this paper reports the finding that *B. subtilis* cells express protease at varying levels and the authors of that paper raise the possibility that protease producing cells act altruistically. In that study cells are cultured in a liquid medium in which they seem to flow more or less freely. If this is the case, then it seems to me that the lifecycle excludes coordinated production of protease cells.

Furthermore, I could not find any indication that cells differ in their level of reproduction and it is therefore unclear to me whether it is justified to classify *B. subtilis* as species with reproductive division of labor.

See above.

Salmonella enterica

Division of labour occurs between self-sacrificing cells that enter the gut tissue and reproducing cells that stay in the gut lumen. To me, it seems likely that the random phenotype determination that is present in this example can again most easily be explained by their function. The two different cell types move independently, self-sacrificing cells enter the gut tissue while reproducing cells stay in the gut lumen, and I cannot envisage how these different activities could be usefully combined with coordinated phenotype determination.

Here, a functional bauplan is likely more easily achieved with random phenotype determination than with coordinated phenotype determination.

See above.

Filamentous cyanobacteria

This case is possibly the most convincing example of a species in which the coordinated determination of phenotypes is partly driven to ensure an optimal ratio of nitrogen to non-nitrogen fixing cells. But in agreement with the authors, I would argue that in this example the optimal spatial arrangement of the two cell types is of even greater importance. This is captured in the new individual-based model presented in the manuscript. However, another aspect of the biology of filamentous cyanobacteria is strangely lacking from this model. My understanding of the literature is that the main explanation for the evolution of two cell types in cyanobacteria is the fact photosynthesis and nitrogen fixation are two physiologically incompatible tasks. Thus, the general view seems to be that division of labour is between nitrogen fixing and photosynthesising cells, not between nitrogen fixing and reproductive cells. Thus, I presume that there is bi-directional exchange of products, photosynthesising cells are also delivering their products to the nitrogen fixing cells. This process is ignored by the authors in their detailed model. If their aim is to make a system specific model, I believe that it has to be included since this ought to be an additional factor avoiding the clumping of nitrogen fixing cells.

The reviewer raises a good point in that cyanobacteria filaments engage in many complex behaviours that we did not formally model in our simulations, including non-reproductive division of labour and bet-hedging. We have altered the text to clarify both the goals and scope of the simulation model and of aspects of cyanobacteria biology that are ignored. Lines 347-350.

The main purpose of the simulation model was to extend our analysis of reproductive division of labour to include the impact of within group spatial structure, where it is not just the proportion of helpers but where they are situated in the group that matters. A more detailed model of cyanobacteria division of labour was an opportunity to investigate the impact of the simplest conceivable group structure in a biologically relevant system.

Focusing only on the reproductive division of labour allows us to connect the results of the simulation model more easily to the main model and to discuss how our results may extend to other systems that employ reproductive division of labour with more complex spatial structures (such as *V. carteri*). Further, we find that coordination is highly favoured in the spatial model (Figure 5E), including details of the non-reproductive division of labour would simply add an additional cost to helpers from being too far from reproductives, which would only make random specialisation even less likely (no reason to expect qualitative change in key predictions).

Dictyostelium discoideum

Division of labor in this example occurs between non-viable stalk cells that hold up and help disperse viable spore cells. This example is similar to *Volvox* in the sense that I believe producing the optimal ratio between reproductive and helper cells is likely not important for the evolution coordinated cell type determination.

Rather, deterministic cell type determination is essential to produce a functional phenotype (stalk and fruiting body).

We disagree that the ratio of helpers to reproductives does not affect the fitness of the fruiting body. If there are not enough stalk cells, then the fruiting body cannot hold up the spores for dispersal (Smith et al. 2014 BMC Evolutionary Biology 14:105). If there are too many stalk cells, this would reduce the number of spores that could be dispersed. These factors would act as an important pressure for coordination to ensure the optimal ratio of stalk to spore cells. The strength of this pressure in our model can be tuned by the relative importance of cooperation (h/b).

In summary, the authors argue for the importance of group size, cost of random and coordinated phenotype determination and importance of cooperation for the evolution of different mechanisms for phenotype determination. However, I believe that the examples given in the manuscript might be better explained by functional constraints.

I like the simplicity of their simple model but it is rather unclear to me whether it has any explanatory power for the standard microbial examples of division of labour. At this stage, I wonder whether system specific models as the one suggested for filamentous cyanobacteria are a more fruitful avenue (although such a model might have to contain additional details).

See above.

Additional comments on the detailed model for filamentous cyanobacteria. From reading the main text it did not become clear to me how is it determined which exact reproductive cell undergoes the next cell division? Is this completely random or does it depend on its internal level of N_2 ?

We have amended the text to make this more clear (Line 339-340).

II. 387-389

Does a biological motivation exist for this simulation algorithm or is it chosen just for convenience?

We have amended the text to better motivate the simulation algorithm (Line 410-412) and have included an additional figure in the supp info to illustrate trait convergence (Supp. Figure 5).

Does the optimal proportion q^* depend on η , the cooperation range?

Yes, as the cooperation range increases, the optimal proportion q^* decreases. This pattern is now present in Figure 5C of the main text (Line 425).

It is not obvious to me that a smaller η is analogous to a smaller social group (smaller n ; line 419). If the optimal q^* is independent of η , then q^* is determined by the total colony ontogeny and maybe the maximal group size. In particular, the sentence "With random specialisation, a smaller social group can lead to proportions of helpers that show greater deviations from the optimum, increasing the benefit that can be obtained by coordination (Figure 3)" would need further explanations.

A change in η has a direct effect on the effective size of the social group. We have amended the text to make this clearer (Lines 441-445). To minimise confusion, we have re-labelled this parameter as a diffusion factor of fixed N_2 rather than the more abstract "cooperation range" (Lines 343-346).

The effect of $\bar{\phi}$ does not seem to be explored in the simulations, and the remark in ll. 434-437 seems therefore a bit unconnected.

We point out here that a change in $\bar{\phi}$ has multiple effects. This can be seen directly from the construction of the model. We do not make any claims about which of these effects is dominant, or whether this would be more or less likely to produce coordination.

Minor points

Throughout the manuscript the use of italics for mathematical symbols is applied inconsequential.

Thank you for pointing this out, we have tried to make sure that all equations are italicised.

The equation at the end of l. 143 is displayed incompletely but I think it is given in Eq. S5 in the Appendix.

Thank you for point this out. A complete mathematical statement would be distracting in the main text and so we have replaced this with a verbal statement. (Lines 148).

l. 159

"sometimes" should be "most of the time". The probability that exactly np^* helpers are produced goes to zero as n goes to infinity.

Amended (Line 164).

l. 193: It is not obvious to me why the expression $(h-b)/n(h+b)$ represents the fecundity

benefits of switching from random specialisation to coordination.

Switching from random specialisation to coordination can only have two effects: a change in the metabolic cost of the mechanism, and an increase in the expected fecundity of reproductives associated with coordination. The relative metabolic cost is entirely captured by the left-hand side of equation 5 and what this balances against must be the relative benefits of coordination.

I. 220 sometimes -> rarely

This sentence has been re-worded (Line 223-226).

Reviewer #2 (Remarks to the Author):

I am pleased to see that authors substantially revised their manuscript on two important fronts. First, the authors provide a more nuanced evaluation of the empirical literature in relation to their model, which is less speculative and does better justice to the scope of their model. Second, besides a number of extensions on their phenomenological model, the authors now also included a more mechanistic model that examines how a division of labor can evolve in Cyanobacteria. This model recapitulates some of the more trivial predictions of the phenomenological model (e.g. coordination is more likely to evolve at low costs and high relative benefits of cooperation) and also reveals some differences, which makes it an interesting addition. Most importantly, the mechanistic model shows that coordination could evolve not only because it leads to the correct fraction of specialized cells, but also because it could lead to an even distribution of those cells. With random specialization, the same fraction of specialized cells leads to reduced benefits of cooperation, because those specialized cells tend to clump more, which effectively reduces the associated benefits. This mechanistic model shows that one should be cautious with generalizing insights obtained from phenomenological models.

Although I am pleased with the revised manuscript, I unfortunately think an additional revision is required. My comments pertain mostly to the newly introduced model on Cyanobacteria that comes with a number of ad hoc assumptions that are insufficiently well motivated and may even require revision.

(1) The authors assume that signal production reduces nitrogen fixation of the helper cells, while signal sensitivity delays timing of replication (modeled through an increased growth cap at division). It is unclear why increased sensitivity would change the timing of division. Would it not be more likely that the costs of sensitivity, which may be caused by increased expression of receptor proteins, result in a reduced growth rate, rather than changing the cell size at division. Why did the authors make this assumption and how are their results different when sensitivity affects growth rate directly?

We have amended the language to be clearer on this point (Lines 388-391).

Determining which phenotype to adopt occurs in only a small window at the beginning of a cell's cycle, and so any costs due to increased sensitivity can only be paid in this window of time. If higher sensitivity incurred a growth-rate cost, then this would act over the entire span of a cell's cycle.

Rather, we assume that more sensitive cells must take longer to determine the exact density of signalling molecule before adopting a phenotype and so there is an increasing time delay before reproductive cells start growing.

Qualitatively, however, we expect this would produce the same broad results as a growth rate cost.

(2) Why do the authors assume that hormogonia give rise to filaments with specialized cells at the tips of the filaments. What happens when authors start from undifferentiated filaments, as observed in Cyanobacteria? Since coordinated spacing requires the presence of specialized cells, I can imagine that starting from undifferentiated filaments necessitates a higher rate of stochastic differentiation.

This is a good point.

We have amended to text to clarify this (Lines 327-330). We assume that group starts with some helpers so that we can look at what happens when there is no fixed N_2 in the environment (essential cooperation).

In addition, we have followed the referee's suggestion and run additional simulations to determine the outcome when groups start as all reproductives. We have amended the main text to say that we found the same qualitative result and include a new section in the supplementary figure (Lines 330-331; linking to supp info figure 6).

(3) As already mentioned above, I think that it is interesting that coordination evolves for a wider range of parameter conditions in the mechanistic model than in the phenomenological model, because there is an additional benefit of an even distribution of specialized cells. Could authors quantify the costs of clumping? Do I understand correctly that clumping is particularly common in the model because cell differentiation is coupled to division? Would you still see pervasive clumping when cell differentiation is uncoupled from division?

This is a great suggestion, which we have followed.

To investigate the impact of helper clumping on the optimal mechanism: we have run additional simulations and have updated our manuscript with: an additional section in the main text (Lines 464-517), a new figure in the main text (Figure 6; Line 500), and an additional supplementary information section (Supp info G.5 and Supplementary Figure 7).

In Figure 6, we show that coordinated specialisation dramatically reduces the average size of helper clumps produced by growing filaments. However, we also find that the potential cost of clumping is greater for coordinated

specialisation as the helpers are already producing less of the public good. This greater cost for coordinated specialisation shows that coordination is favoured because it reduces the probability of forming clumps, not because clumps are less costly. In the supplementary Figure 7, we also show that random specialisation in a filament that doesn't grow (uncoupling division and differentiation) can still produce clumps, but with a smaller average clump size.

(4) It is unclear why authors first let the differentiation probability (q) evolve for 500 generations and then the remaining parameters underlying coordination. Why do these parameters not evolve simultaneously? If q can evolve simultaneously with the other parameters, would the parameter space for which coordination evolves become larger or smaller?

We have amended the main text to make this clearer (410-412) and include an additional figure in the supp info to illustrate the convergence of our simulations (Supplementary Figure 5).

(5) I would suggest the authors to show all four evolving parameters in Figure 5?

Amended (Line 425).

(6) The authors say "Colonies of *Volvox carteri* and *Dictyostelium discoideum* use coordination to divide labour, despite the fact that these groups are composed of large numbers of cells (high n ; on the order of 1000s of cells or more). However, the initial specialisation of helpers in each case appears to make use of a shared developmental programme and so the relative costs of coordination could be quite small for both species (low γ)"

I do not understand how authors can draw this conclusion. How is the developmental programme shared by cells in *Dictyostelium* or *Volvox* different from that shared by cells within a filament in *Cyanobacteria*? I consider it way more plausible that the phenomenological model, as presented by the authors, simply fails to capture all the benefits that could drive the evolution of coordination. In the phenomenological model, coordination provides a single benefit only, by leading to the correct fraction of specialized cells. However, as the authors also show with their model on *Cyanobacteria*, other benefits might follow from coordination – such as an even distribution of specialized cells. The phenomenological model might simply fail to capture such benefits, leaving it unexplained why coordination could also evolve in larger groups. I would strongly encourage the authors to better highlight the limitations of their phenomenological model.

Thank you for the helpful feedback. We have updated our discussion to reflect these points (Lines 536-546).

(7) The authors continue by saying that "Further, in both cases an absence of helpers would lead to cell death and thus cooperation is important (higher h/b)"

As pointed out previously, here again the authors risk tautological reasoning: specialization evolved, because specialization is essential, but specialization might as well have become essential after it first evolved (from a non-essential state). In fact, the latter is much more plausible for Dictyostelium, as the ancestor of Dictyostelium was hypothesized to be capable of surviving starvation through encystation (which is widely conserved among the solitary and many social slime molds), from which fruiting body formation and sporulation later evolved. By the same token, in some species of Cyanobacteria, knocking out heterocyst differentiation would lead to cell death in the absence of a nitrogen source, but in many other species, there are no specialized cells, because vegetative cells simply cycle between photosynthesis and nitrogen fixation.

Amended (Lines 536-546).

Reviewers' Comments:

Reviewer #1:

Remarks to the Author:

In line 159 the authors state "We assume that the probability of adopting a helper role is equal to the optimal proportion of helpers ($p^* = k^* / n$)."

Given that the authors use an optimisation argument this assumption hinges implicitly on the assumption that p^* maximises w_R . Given that w_R results from summing over a binomial distribution where each realisation is weighted with $g_{k,n}$ I wonder whether this implicit assumption is actually true? For example, I would anticipate that a p -value somewhat higher than p^* could maximise w_R when groups that have a higher proportion of helpers than p^* have a higher w_R than groups that have a lower proportion of helpers than p^* .

More generally, I wonder whether the authors could prove under which conditions their assumption is or is not true? If such a proof is not achievable, then the authors still might want to discuss the implications of this assumption not applying. I envisage that when this assumption does not apply inequality (5) might no longer hold.

I. 543-553

I still would like to suggest that the current frameworks largely ignores an important factor that one might want to term the "functional costs of random and coordinated specialisation". I find it likely that such functional costs can be an important selective force in the evolution of random vs coordinated specialisation in many examples of evolved division of labor, and, in my opinion, functional costs are not captured by the general framework presented here, which emphasises the physiological costs of random and coordinated specialisation, the effect of group size on producing the optimal ratio of reproductive and helper cells, and the importance of helping.

In this paragraph, the authors now allude to these functional costs in connection with *Volvox*: random phenotype specialisation cannot produce a functional colony in which flagella bearing cells form a hull around reproductive cells in the interior.

Similarly, I would like to argue that in the case *Dictyostelium* we observe coordinated specialisation because random specialisation cannot produce a functional phenotype, i.e., a phenotype consisting of a stalk and a fruiting body. I find it plausible that it is this functional cost that is the main driver for coordinated specialisation rather than any of the factors investigated by the authors in their general model.

I. 119

"Throughout, we assume a form of cooperation that is common in microbes, ..."

Why not add references supporting this statement?

I. 136

"often"

Could this be replaced with a less vague statement?

I. 238

"in many scenarios"

Again, could this be replaced with a less vague statement?

I. 252

I don't find it obvious on what the "Consequently" is based here. Why is obvious that the fact that in smaller groups fewer possible outcomes for the proportion of helpers exist can more easily lead to the formation of groups with a realised proportion of helpers that deviates significantly from the optimum? It seems to me that the true reason is given in the next paragraph (law of large numbers).

I. 429

Maybe replace "suggest" with "indicate"?

Reviewer #2:

Remarks to the Author:

I thank the authors for their thoughtful revision. They have sufficiently addressed my concerns by running additional simulations, expanding their results and revising parts of the discussion. I would therefore now support publication.

Although the authors have sufficiently addressed my concerns, while reading the review replies, I noticed that the authors made several mistakes in their rebuttal to the first reviewer, who – to my opinion – raises some valid concerns. For instance, in the main text, the authors use *Bacillus subtilis*' heterogeneous protease production as an example of a reproductive division of labor. Reviewer 1 correctly points out that there is no proof for whether heterogeneous protease production actually concerns a reproductive division of labor. In their rebuttal, the authors cite a list of papers, but only one of the cited papers actually examines protease production (Veening et al. 2008, *Molecular systems biology*), the others do not. In the paper of Veening et al., 2008, there is no quantification of relative division rates, which makes it impossible to conclude whether heterogeneous protease production concerns a reproductive division of labor. Some of the other concerns of reviewer 1 are simply not addressed (e.g. the comment regarding *Salmonella*'s division of labor), while I think they are fair. I think reviewer 1 correctly points out that the model does not provide a general explanation for why random or coordination division of labor evolved in microbes, because many potentially important factors are not addressed in the model. In the revised manuscript, the authors do acknowledge this limitation of their model in the discussion.

Dear Wenfei and referees,

Please find below our responses to the referee comments, whom we thank for their time and help in improving the manuscript.

All the best,

Guy + co-authors

REVIEWER COMMENTS

Reviewer #1 (Remarks to the Author):

In line 159 the authors state "We assume that the probability of adopting a helper role is equal to the optimal proportion of helpers ($p^* = k^* / n$)."

Given that the authors use an optimisation argument this assumption hinges implicitly on the assumption that p^* maximises w_R . Given that w_R results from summing over a binomial distribution where each realisation is weighted with $g_{k,n}$ I wonder whether this implicit assumption is actually true? For example, I would anticipate that a p -value somewhat higher than p^* could maximise w_R when groups that have a higher proportion of helpers than p^* have a higher w_R than groups that have a lower proportion of helpers than p^* .

More generally, I wonder whether the authors could prove under which conditions their assumption is or is not true? If such a proof is not achievable, then the authors still might want to discuss the implications of this assumption not applying. I envisage that when this assumption does not apply inequality (5) might no longer hold.

This is a very good point. Our submission did in fact include an analysis of this exact scenario in Supp. Info. D.2, and we found that there were no qualitative differences in the predictions of our model when accounting for the possibility that random specialisers maximise their own p -value. We have added a reference to this analysis in lines 303-304 of the main text.

I. 543-553

I still would like to suggest that the current frameworks largely ignores an important factor that one might want to term the "functional costs of random and coordinated specialisation". I find it likely that such functional costs can be an important selective force in the evolution of random vs coordinated specialisation in many examples of evolved division of labor, and, in my opinion, functional costs are not captured by the general framework presented here, which emphasises the physiological costs of random and coordinated specialisation, the effect of group size on producing the optimal ratio of reproductive and helper cells, and the importance of helping.

In this paragraph, the authors now allude to these functional costs in connection with

Volvox: random phenotype specialisation cannot produce a functional colony in which flagella bearing cells form a hull around reproductive cells in the interior.

This is a good point that the costs of each mechanism can come from multiple sources. The costs in our model are not just metabolic/physiological – they can take multiple forms. We allow for both a fixed metabolic cost to each mechanism (the c terms) and a functional cost from producing a suboptimal proportion of helpers (how fitness depends upon group composition), which we previously referred to as the “stochastic cost”.

To emphasise this distinction, we have streamlined terminology and clarified the main text (lines 150-152, 165-169, 207-208, 243-244, 284-286, 479), and added a more detailed discussion of costs and functional constraints to Supp. Info H.

Similarly, I would like to argue that in the case Dictyostelium we observe coordinated specialisation because random specialisation cannot produce a functional phenotype, i.e., a phenotype consisting of a stalk and a fruiting body. I find it plausible that it is this functional cost that is the main driver for coordinated specialisation rather than any of the factors investigated by the authors in their general model.

Random specialisation could in principle occur in this example. We clarify this in a new discussion section in Supp. Info. H.

I. 119

“Throughout, we assume a form of cooperation that is common in microbes, ...”
Why not add references supporting this statement?

We have added references, including papers for specific cases and across species reviews of microbial cooperation. Line 118.

I. 136

“often”
Could this be replaced with a less vague statement?

We have amended the text. Line 134-136.

I. 238

“in many scenarios”
Again, could this be replaced with a less vague statement?

Text amended. Line 237-238.

I. 252

I don't find it obvious on what the “Consequently” is based here. Why is obvious that the fact that in smaller groups fewer possible outcomes for the proportion of helpers exist can more easily lead to the formation of groups with a realised proportion of helpers

that deviates significantly from the optimum? It seems to me that the true reason is given in the next paragraph (law of large numbers).

We have truncated and merged this paragraph and the next to emphasise the true reason for the pattern as suggested by the reviewer. Lines 257-265.

I. 429

Maybe replace "suggest" with "indicate"?

Amended. Line 431.

Reviewer #2 (Remarks to the Author):

I thank the authors for their thoughtful revision. They have sufficiently addressed my concerns by running additional simulations, expanding their results and revising parts of the discussion. I would therefore now support publication.

We thank the reviewer for their helpful suggestions and feedback.

Although the authors have sufficiently addressed my concerns, while reading the review replies, I noticed that the authors made several mistakes in their rebuttal to the first reviewer, who – to my opinion – raises some valid concerns. For instance, in the main text, the authors use *Bacillus subtilis*' heterogeneous protease production as an example of a reproductive division of labor. Reviewer 1 correctly points out that there is no proof for whether heterogeneous protease production actually concerns a reproductive division of labor. In their rebuttal, the authors cite a list of papers, but only one of the cited papers actually examines protease production (Veening et al. 2008, *Molecular systems biology*), the others do not. In the paper of Veening et al., 2008, there is no quantification of relative division rates, which makes it impossible to conclude whether heterogeneous protease production concerns a reproductive division of labor.

We have added a sentence in the main text to reflect further experimental work required in the *B. subtilis* system. Line 185-186.

Some of the other concerns of reviewer

1 are simply not addressed (e.g. the comment regarding *Salmonella*'s division of labor), while I think they are fair. I think reviewer 1 correctly points out that the model does not provide a general explanation for why random or coordination division of labor evolved in microbes, because many potentially important factors are not addressed in the model. In the revised manuscript, the authors do acknowledge this limitation of their model in the discussion.

We address this point in the discussion of the main text (Lines 550-553) and in a new section of the Supp. Info. (section H).

Reviewers' Comments:

Reviewer #1:

Remarks to the Author:

In response to the first comment from my previous review the authors write "We have added a reference to this analysis in lines 303-304 of the main text."

The authors indeed changed the text in lines 303-304, I can however not see any reference to the scenario analysed in Appendix D.2 nor an explicit reference to that Appendix. I also think it would be more appropriate to add such a reference around line 162.

In my previous reviews I suggested (and this suggestion seems to have been supported by the second reviewer) that both coordinated and random specialisation might not be able to produce a "functional phenotype". I suggested the term "functional costs" to refer to the idea that both types of phenotype determination can come with constraints in terms of the e.g. spatial arrangement of the different cell types that might not be well suited for the different cell types to execute their task efficiently. The authors now refer in several places in their manuscript to this possibility. To my surprise, however, they have now opted to label the right-hand side of inequality (5) as functional costs (l. 239-240 and elsewhere). I have two comments. First, the text in the main part contains not explanation as to why the right-hand side of inequality (5) can be interpreted as a cost arising from the stochasticity in stochastic phenotype determination. This connection is only mentioned in passing in line 239. Second, I don't find "functional costs" an obvious choice as a term for this type of cost (and it is certainly not how I suggested to use the term). I think that the previous term "stochastic cost" was more suitable since it more directly referred to the cause of the cost.

In line 123 (and other places) the authors refer to $g_{\{k, n\}}$ as group fecundity (the reproductive success of a particular group in the absence of mechanism costs) and go on by writing that it is the per capita number of offspring that would disperse at the end of the group life cycle. I find it confusing to refer to a per capita quantity as group fecundity. Maybe in the context of the manuscript it is also preferable to use per-cell instead of per-capita.

l. 363

Here the authors made an addition to the sentence, which now reads "We allowed for reproductive fecundity to depend non-linearly on the proportion of helpers in the group, for random specialisers to maximise their own probability of becoming helpers (Equation 3), ..."

Maybe better: for randomly specialising cells to maximise their own probability of becoming helpers.

In the context of this paper, I interpret a "random specialiser" as being a species or genotype but not a cell.

Reviewer #1 (Remarks to the Author):

In response to the first comment from my previous review the authors write "We have added a reference to this analysis in lines 303-304 of the main text."

The authors indeed changed the text in lines 303-304, I can however not see any reference to the scenario analysed in Appendix D.2 nor an explicit reference to that Appendix. I also think it would be more appropriate to add such a reference around line 162.

We have added a brief motivation and explicit reference to appendix D.2 (now supplementary method A.2) in the main text at lines (lines 223-228).

In my previous reviews I suggested (and this suggestion seems to have been supported by the second reviewer) that both coordinated and random specialisation might not be able to produce a "functional phenotype". I suggested the term "functional costs" to refer to the idea that both types of phenotype determination can come with constraints in terms of the e.g. spatial arrangement of the different cell types that might not be well suited for the different cell types to execute their task efficiently. The authors now refer in several places in their manuscript to this possibility. To my surprise, however, they have now opted to label the right-hand side of inequality (5) as functional costs (l. 239-240 and elsewhere). I have two comments. First, the text in the main part contains not explanation as to why the right-hand side of inequality (5) can be interpreted as a cost arising from the stochasticity in stochastic phenotype determination. This connection is only mentioned in passing in line 239. Second, I don't find "functional costs" an obvious choice as a term for this type of cost (and it is certainly not how I suggested to use the term). I think that the previous term "stochastic cost" was more suitable since it more directly referred to the cause of the cost.

Where appropriate have returned the terminology to refer to stochastic costs of random specialisation rather than functional costs (lines 233, 234, 277-283, 334, 335, 377, 770-771).

We have added an explanation for terming the right-hand side of Equation 5 as the stochastic cost (lines 277-283).

In line 123 (and other places) the authors refer to $g_{\{k, n\}}$ as group fecundity (the reproductive success of a particular group in the absence of mechanism costs) and go on by writing that it is the per capita number of offspring that would disperse at the end of the group life cycle. I find it confusing to refer to a per capita quantity as group fecundity. Maybe in the context of the manuscript it is also preferable to use per-cell instead of per-capita.

We now refer to this quantity as the average group fecundity and use per-cell rather than per-capita. (lines 185, 210, 826)

I. 363

Here the authors made an addition to the sentence, which now reads "We allowed for reproductive fecundity to depend non-linearly on the proportion of helpers in the group, for random specialisers to maximise their own probability of becoming helpers (Equation 3), ..."

Maybe better: for randomly specialising cells to maximise their own probability of becoming helpers.

In the context of this paper, I interpret a "random specialiser" as being a species or genotype but not a cell.

Amended (lines 400-401).